# Diversification of dentate gyrus granule cell subtypes is regulated by Nrg1 nuclear back-signaling

Prithviraj Rajebhosale[1] , Li Jiang[1] , Haylee J Ressa[2] , Kory R Johnson[3], Niraj S Desai[4] , Alice Jone[5], Lorna W Role[4], David A Talmage[1]

**Neuronal heterogeneity is a defining feature of the developing mammalian brain, but the mechanisms regulating the diversification of closely related cell types remain elusive. Here, we investigated granule cell (GC) subtype composition in the dentate gyrus (DG) and the influence of a psychosis-associated $V_{321}L$ mutation in Neuregulin1 (Nrg1). Using morphoelectric characterization, single-nucleus gene expression, and chromatin accessibility profiling, we identified distinctions between typical GCs and a rare subtype known as semilunar granule cells (SGCs). We found that the $V_{321}L$ mutation, which disrupts Nrg1 nuclear back-signaling, results in overabundance of SGC-like cells. Pseudotime analyses suggest a GC-to-SGC transition potential, supported by the accessibility of SGC-enriched genes in non-SGCs. In WT mice, SGC-like gene expression increases during adolescence, coinciding with reduced Nrg1 back-signaling capacity. These results suggest that intact Nrg1 nuclear signaling represses SGC-like fate and that its developmental or pathological loss may permit acquisition of this fate. Our findings reveal a novel role of Nrg1 in maintaining DG cell-type composition and suggest that disrupted subtype regulation may contribute to disease-associated changes in the DG.**

## Introduction

Heterogeneity of neuronal cells is an emergent property of the developing mammalian brain (Tasic et al, 2016; Cembrowski & Spruston, 2019; BRAIN Initiative Cell Census Network, 2021; La Manno et al, 2021; Yao et al, 2023). However, how closely related neuronal cell types diversify remains mechanistically elusive (Morris, 2019; Zeng, 2022). In this study, we comprehensively profiled differences between two closely related, yet morphologically distinct, dentate gyrus (DG) granule cell (GC) types—typical GCs and

semilunar granule cells (SGCs)—to investigate molecular determinants of individual cell-type identity.

Generation of GCs in the DG proceeds in two main phases—embryonic and postnatal, which, in mice, is further divided into a perinatal burst of neurogenesis (P0–P8) and "adult" neurogenesis (P8 into adulthood). Embryonically, GCs are produced from neural precursors (NPs) located in the dentate neuroepithelium (Hatami et al, 2018). These precursors migrate to establish a new neurogenic niche below the granule cell layer (GCL) known as the subgranular zone (SGZ), which gives rise to new GCs throughout life. Detailed transcriptomic analyses have concluded that the gene expression profiles of the dentate neuroepithelium and SGZ NPs do not significantly differ and that the neurogenic trajectories, that is, sequential changes in the transcriptome as neurons differentiate, are also highly conserved between embryonic and postnatal development (Hochgerner et al, 2018; Berg et al, 2019). However, a notable difference between embryonically and postnatally born GCs is that about half of the GCs produced during embryonic development are morphologically unique, bearing multiple splaying primary dendrites, known as SGCs, which eventually only comprise up to 3% of all DG GCs (Save et al, 2019). SGCs are not generated postnatally as evidenced by a spatial bias in their location within the GCL; SGCs are predominantly located in the suprapyramidal blade of the DG where they populate the dorsalmost layers bordering the molecular layer (MOL) along with morphologically typical GCs, which are born around the same time. Postnatally born GCs populate the GCL in an "outside-in" manner and are in cell layers closer to the hilus of the DG (Save et al, 2019). Studies characterizing the dendritic morphology of GCs and SGCs have concluded that the differences between these cells exist as early as 1–2 wk after birth and persist thereafter, concluding that the SGC phenotype represents a bona fide cell type, supporting the conclusion that the two GC subtypes are fundamentally distinct cell types (Gupta et al, 2020).

The significance of understanding mechanisms related to heterogeneity of GCs is underscored by numerous findings of

[1]Genetics of Neuronal Signaling Section, National Institute of Neurological Disorders and Stroke, NIH, Bethesda, MD, USA   [2]Undergraduate Program in Fundamental Neuroscience, University of Virginia, Charlottesville, VA, USA   [3]Bioinformatics Core, National Institute of Neurological Disorders and Stroke, National Institutes of Health, Bethesda, MD, USA   [4]Circuits, Synapses and Molecular Signaling Section, National Institute of Neurological Disorders and Stroke, NIH, Bethesda, MD, USA   [5]Program in Neuroscience, State University of New York at Stony Brook, Stony Brook, NY, USA

Correspondence: david.talmage@nih.gov
Alice Jone's present address is Regulatory Affairs Division, STERIS Corporation, Mentor, OH, USA

heterotopic and overabundant adult-born neurons with SGC-like morphologies in the DG of mice harboring disease-associated genetic mutations (Kim et al, 2009; Llorens-Martin et al, 2014). A similar phenotype has also been observed in postmortem brains of humans with schizophrenia, Alzheimer's disease, and fronto-temporal dementia (Lauer et al, 2003; Senitz & Beckmann, 2003; Terreros-Roncal et al, 2019; Marquez-Valadez et al, 2022). These data imply potential conversion of GCs to SGCs or, at the very least, a preserved capacity of the adult DG to generate SGC-like cells. In addition, implicit in these findings is the idea that dysregulation of the cell biological mechanisms regulating subtype specification or maintenance might underlie the altered composition of the GC population in disease states.

In this study, we performed morphoelectric and molecular characterization of GCs and SGCs to reveal cell type–specific features that distinguish these granule cell subtypes. Trajectory analyses on single-nucleus gene expression and chromatin accessibility data revealed a potential for a GC→SGC genetic conversion, which has been implicated in aging and genetic associations with psychiatric disease. In line with this, we found that a psychosis-associated missense mutation (Val$_{321}$→Leu; V$_{321}$L) in Neuregulin1 (*Nrg1*) resulted in overabundant heterotopic SGCs and that adolescence-to-adulthood transition in the WT DG was accompanied by increased numbers of cells with SGC-like gene expression signatures. The V$_{321}$L mutation diminishes the ability of Nrg1 proteins to perform membrane-to-nucleus signaling (nuclear back-signaling), which was shown to regulate adult neurogenesis in the DG (Rajebhosale et al, 2024). We found that the expression of requisite components of the Nrg1 nuclear back-signaling pathway decreases over age. Thus, we conclude that intact Nrg1 nuclear back-signaling suppresses the SGC-like fate, and loss of this signaling results in derepression of the SGC-specific gene expression signature resulting in acquisition of SGC-like properties.

# Results

## SGCs have unique morphological and electrical properties

Using Golgi staining, we identified two morphologically distinct types of GCs distinguished by the number of primary dendrites (Fig 1A). Previously published reports have characterized a rare GC-like population known as SGCs, which bear multiple splaying primary dendrites with cell bodies located near or within the inner molecular layer of the DG, and unique electrical properties (y Cajal, 1911; Williams et al, 2007; Save et al, 2019; Gupta et al, 2020). Given the limitations of unambiguously resolving cell position within the GCL using Golgi staining, we performed whole-cell patch clamp recordings from acute hippocampal slices with biocytin fills to reconstruct morphologies of recorded neurons, while simultaneously obtaining information regarding their intrinsic electrical properties (Fig 1B and C). We recorded GCs from the middle of the supra-pyramidal blade of the DG and avoided the lower cell layer to avoid immature GCs generated from neural stem cells located in the SGZ. Similar to our results from Golgi staining, we found recorded and

filled neurons with single and multiple primary dendrites that we classified as GCs and SGCs, respectively (Fig 1B–E).

In the DG of WT mice, we found that SGCs were in the top cell body layers (<20 $\mu$m from the surface) and cells with the typical GC morphology were found throughout the GCL (Fig 1F, $P < 0.0001$). SGCs had wider dendritic splay angles (Fig 1G, $P = 0.02$) and less complex dendrites (Fig 1H, branch points per dendrite, $P = 0.002$; Fig 1I, length per dendrite, $P = 0.0004$). These findings match existing reports regarding morphologies of embryonically born GCs (Kerloch et al, 2019).

We recorded the membrane voltage responses of GCs and SGCs to current steps. Action potentials (APs) at rheobase were analyzed for 20 features. We found that SGCs had a significantly hyper-polarized resting membrane potential (RMP) (Fig 1J, $P = 0.02$), shorter AP width (Fig 1K, $P = 0.01$), and faster downstroke (Fig 1L, $P = 0.046$). The pronounced differences in the kinetics of the SGC action potentials can be seen in the phase plots for APs at rheobase (Fig 1M).

Because SGCs are born earlier than most other GCs, we asked if the group differences between GCs and SGCs could be reflective of cellular age as opposed to cell type–defining features. Given the stereotyped relationship between GCL lamination and neuronal birthdate in the DG, we analyzed the relationships between the measured electrical properties and soma position within the GCL. We found that RMP (Fig S1A, $P = 0.0129$), spike width (Fig S1B, $P = 0.0543$), and downstroke (Fig S1C, $P = 0.0139$) significantly covaried with soma position as expected from the GC versus SGC group analyses (Fig 1J–L). We also found that rheobase (Fig S1D, $P = 0.0475$), sag (Fig S1E, $P = 0.0359$), and upstroke–downstroke ratio (Fig S1F, $P = 0.0131$) significantly covaried with soma positions in the GCL. The remainder of the properties did not differ significantly between GCs and SGCs or by soma position.

Next, we assessed the correlations between these six properties and soma positions for GCs alone (excluding SGCs) to identify electrical properties that might reflect effects of cell birthdate rather than the cell type. Of the six properties, we found that downstroke and sag were still significantly correlated with soma positions (Fig S1I, downstroke, $P = 0.006$, and Fig S1K, sag, $P = 0.03$), whereas the rest of the properties did not correlate with soma positions (Fig S1G, RMP, $P = 0.22$, Fig S1H, spike width, $P = 0.1$, Fig S1J, rheobase, $P = 0.06$, and Fig S1L, upstroke–downstroke ratio, $P = 0.4$). These results indicate that the differences between GCs and SGCs in RMP and spike width are cell type–defining features.

Thus, our findings define electrophysiological differences between GCs and SGCs further substantiating published reports of electrophysiological differences between GCs and SGCs (Williams et al, 2007; Kerloch et al, 2019).

## V$_{321}$L mutation in Nrg1 results in altered composition of the DG granule cell population

Cells with SGC-like morphologies are overabundant in postmortem brains of people with neuropsychiatric conditions and in rodent experimental systems modeling psychiatric disease–associated genetic and environmental factors (Lauer et al, 2003; Senitz & Beckmann, 2003; Kim et al, 2009; Fitzsimons et al, 2013; Llorens-Martin et al, 2014; Howe et al, 2017; Terreros-Roncal et al, 2019; Marquez-Valadez et al, 2022). We previously characterized a

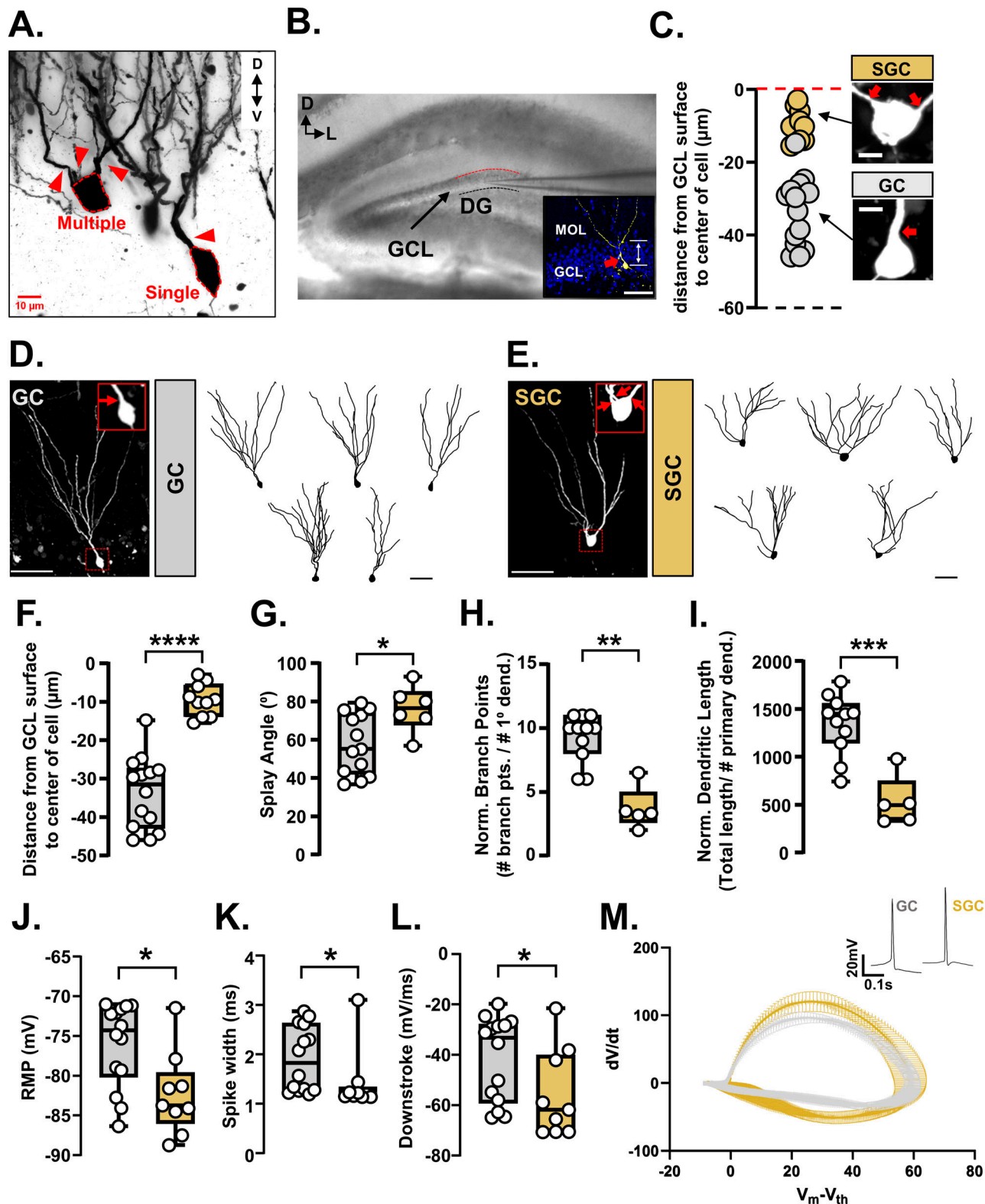

**Figure 1. Morphological and electrical properties of semilunar granule cells are distinct from granule cells.**
**(A)** Representative image of a WT Golgi-stained DG GCL showing two GCs, one example of a GC with a single primary dendrite and another of a GC with multiple primary dendrites. The cell body is outlined with a red dashed line. Primary dendrites are indicated by red arrowheads. Scale bar = 10 $\mu$m. **(B)** DIC image of a coronal hippocampal section prepared for acute slice electrophysiology. The anatomical legend in the top left corner indicates the orientation of the slice—D = dorsal and L = lateral. The dorsal edge of the granule cell layer (GCL) is demarcated by a red dashed line, and the ventral border by a black dashed line. The internal solution within the patch pipette

psychosis-associated missense mutation in the *NRG1* gene showing that it resulted in reduced stem cell proliferation, increased neuronal differentiation, and altered dendritic branching of GCs in the DG (Rajebhosale et al, 2024). During this study, we qualitatively noted numerous GCs with multiple primary dendrites resembling SGCs in the $V_{321}L$ DG. To quantify the numbers of GCs and SGCs, we used the rapid Golgi staining technique and counted stained neurons with single or multiple primary dendrites from WT and $V_{321}L$ DG (Fig 2A). There was no difference in the total number of cells labeled by the Golgi stain between genotypes (Fig 2A, left; $P$ = 0.68). However, $V_{321}L$ mice had a significantly higher number of Golgi-stained GCs with multiple primary dendrites compared with WT mice and a significantly lower number of GCs with single primary dendrites compared with WT mice (Fig 2A, right; $P$ = 0.011, post hoc comparisons: WT multiple versus $V_{321}L$ multiple, $P$ = 0.008; WT single versus $V_{321}L$ single, $P$ = 0.008). The number of GCs with multiple primary dendrites also significantly differed from the numbers of GCs with single primary dendrites in the $V_{321}L$ DG ($V_{321}L$ single versus $V_{321}L$ multiple, $P$ = 0.013), whereas there was no difference in this regard in WT mice ($P$ = 0.85). These results indicate that $V_{321}L$ mice have a higher number of cells with SGC-like morphologies in the DG, potentially at the expense of the other subpopulation of GCs.

Next, we profiled GCs from $V_{321}L$ mice throughout the GCL for their electrical and morphological properties using whole-cell patch clamp with dye fills. Like the WT DG, we found SGCs in superficial layers and GCs throughout the GCL. However, we also found cells with multiple primary dendrites throughout the GCL in the $V_{321}L$ DG (Fig 2B). Analyzing the locations of all recorded cells, we found no difference between GC and SGC localization (Fig 2C, $P$ = 0.9). There were no significant differences in the splay angle between GCs and SGCs (Fig 2D, $P$ = 0.08). In contrast, the differences in dendritic complexity between GCs and SGCs were preserved in the $V_{321}L$ mutant DG (Fig 2E, normalized branch points, $P$ = 0.0013; Fig 2F, normalized dendritic length, $P$ = 0.0005). To further assess any morphological changes in the $V_{321}L$ DG, we subjected the 15 measured morphological parameters from WT and $V_{321}L$ cells to

classification using linear discriminant analysis (LDA). LDA resulted in successful capture of cell-type differences between GCs and SGCs (along the primary classifier—LD1) but not between genotypes (Fig 2G). Analysis of the weighted contributions of each morphological parameter to the LDs revealed that separation along LD1 was heavily influenced by dendritic complexity parameters in line with the known lower complexity of SGC dendrites (Fig S2A). We also performed unsupervised clustering using k-means clustering, which also resulted in clustering of cells primarily based on the cell type rather than the genotype (Fig S2B). We did note a differential spread of WT and $V_{321}L$ SGCs along LD2, which might indicate a potential change in variance within each dataset; however, given the poor functionality of LD2 projection as a classifier, these data indicate that although the SGC-like cells were overrepresented and mislocalized in the $V_{321}L$, morphological distinctions between GCs and SGCs were preserved (Fig 2H).

We profiled sub- and suprathreshold intrinsic electrical properties of cells in the $V_{321}L$ DG. Unlike WT GCs and SGCs, we found no differences in electrical properties between $V_{321}L$ GCs and SGCs (Fig 2I–K, RMP, $P$ = 0.9; spike width, $P$ = 0.3; downstroke, $P$ = 0.5). The phase plots for APs at rheobase further demonstrate this point showing a virtually complete overlap between GCs and SGCs (Fig 2L).

To further evaluate effects of the cell type and genotype, we performed LDA and unbiased clustering using k-means clustering using all recorded features from WT and $V_{321}L$ neurons. LDA resulted in separation of WT GCs and all other groups (WT SGCs, $V_{321}L$ GCs, and $V_{321}L$ SGCs) along LD1 driven strongly by spike width (Figs 2M and S2C). k-means clustering also showed that $V_{321}L$ GCs and SGCs showed a nearly similar distribution as WT SGCs (Fig S2D). These data indicate that in the $V_{321}L$ mutant DG, SGCs are morphoelectrically "normal" and that morphologically typical GCs assume electrical properties of SGCs.

$V_{321}L$ mice exhibited mislocalized SGCs (Fig 2H). In WT mice, several electrophysiological properties covaried with soma position (Fig S1). To account for both soma position and cell type, we applied a multiple linear regression model to WT mice, then

---

contained biocytin to fill the recorded cells. Scale bar = 100 $\mu$m. DG = dentate gyrus; GCL = granule cell layer. (Inset) A representative confocal image of a patched granule cell filled with biocytin identified via fluorescently conjugated streptavidin. The image shows a maximum z-projection of three 1-$\mu$m optical sections containing the widest portion of the soma used to quantify cell position in the GCL relative to the GCL-MOL boundary (red dashed line). Scale bar = 50 $\mu$m. MOL = molecular layer. **(C)** Quantified cell positions of all recorded GCs from WT mice. The red and black dashed lines correspond to the dorsal and ventral borders of the GCL, respectively. Semilunar granule cells (SGCs—GCs with multiple primary dendrites) were found in the top cell body layer of the GCL (gold circles). Typical GCs with a single primary dendrite were found throughout the GCL (silver circles). Representative images of a filled and recorded SGC and GC are shown. Red arrows point to primary dendrites extending from the cell body. N = 14 cells (GC) and 10 cells (SGC) from 5 mice (20 slices). Scale bar = 10 $\mu$m. **(D)** (Left) Representative image of a filled GC. A dotted red box indicates the region magnified in the inset. Inset shows a magnification of the cell body and proximal dendrite of the filled GCs. The red arrow marks the primary dendrite emanating from the soma. Scale bar = 50 $\mu$m. (Right) Five representative 2-D tracings of filled GCs. Scale bar = 50 $\mu$m. **(E)** (Left) Representative image of a filled SGC. A dotted red box indicates the region magnified in the inset. Inset shows a magnification of the cell body and proximal dendrites of the filled SGCs. Red arrows mark primary dendrites emanating from the soma. Scale bar = 50 $\mu$m. (Right) Five representative 2-D tracings of filled GCs. Scale bar = 50 $\mu$m. **(F)** Quantification of cell body position within the GCL of GCs (silver) and SGCs (gold). SGC cell body locations were significantly different from those of GCs (****$P$ < 0.0001, Welch's $t$ test [two-tailed]; t = 8.471, df = 19.16). N = 14 cells (GC) and 10 cells (SGC) from 5 mice. **(G)** Quantification of splay angle of GCs and SGCs. The splay angle (angle between the two farthest dendrites measured at the center of the cell body) was significantly larger in SGCs compared with GCs (*$P$ = 0.0168, Welch's $t$ test [two-tailed]; t = 2.754, df = 12.59). N = 13 cells (GC) and 6 cells (SGC) from 4 mice. **(H)** Quantification of dendritic branching of GCs and SGCs normalized to the number of primary dendrites. The number of branch points was significantly higher in GCs compared with SGCs (**$P$ = 0.0023, Mann–Whitney test, U = 2). N = 11 cells (GC) and 5 cells (SGC) from 4 mice. **(I)** Quantification of dendritic length of GCs and SGCs normalized to the number of primary dendrites. The normalized dendritic length was significantly higher in GCs compared with SGCs (***$P$ = 0.0004, Welch's $t$ test [two-tailed]; t = 5.348, df = 9.366). N = 11 cells (GC) and 5 cells (SGC) from 4 mice. **(J)** SGCs have a significantly more hyperpolarized resting membrane potential compared with GCs (*$P$ = 0.02, Mann–Whitney test, U = 26; N = 14 cells [GCs] and 9 cells [SGCs] from 5 mice). **(K)** SGCs have a significantly narrower spike width compared with GCs (*$P$ = 0.012, Mann–Whitney test, U = 24; N = 14 cells [GCs] and 9 cells [SGCs] from 5 mice). **(L)** SGCs have a significantly faster downstroke compared with GCs (*$P$ = 0.046, Mann–Whitney test, U = 31; N = 14 cells [GCs] and 9 cells [SGCs] from 5 mice). **(M)** Phase plots of action potential dynamics (dV/dt versus $V_m$-$V_{th}$) provide a visualization of differences in the AP waveform between GCs and SGCs. N = 14 cells (GC) and 9 cells (SGC) from 5 mice. Note: the difference in the number of GCs and SGCs for different morphological and electrical measures was because of incomplete total reconstructions, which precluded inclusion of dendritic morphology parameters.

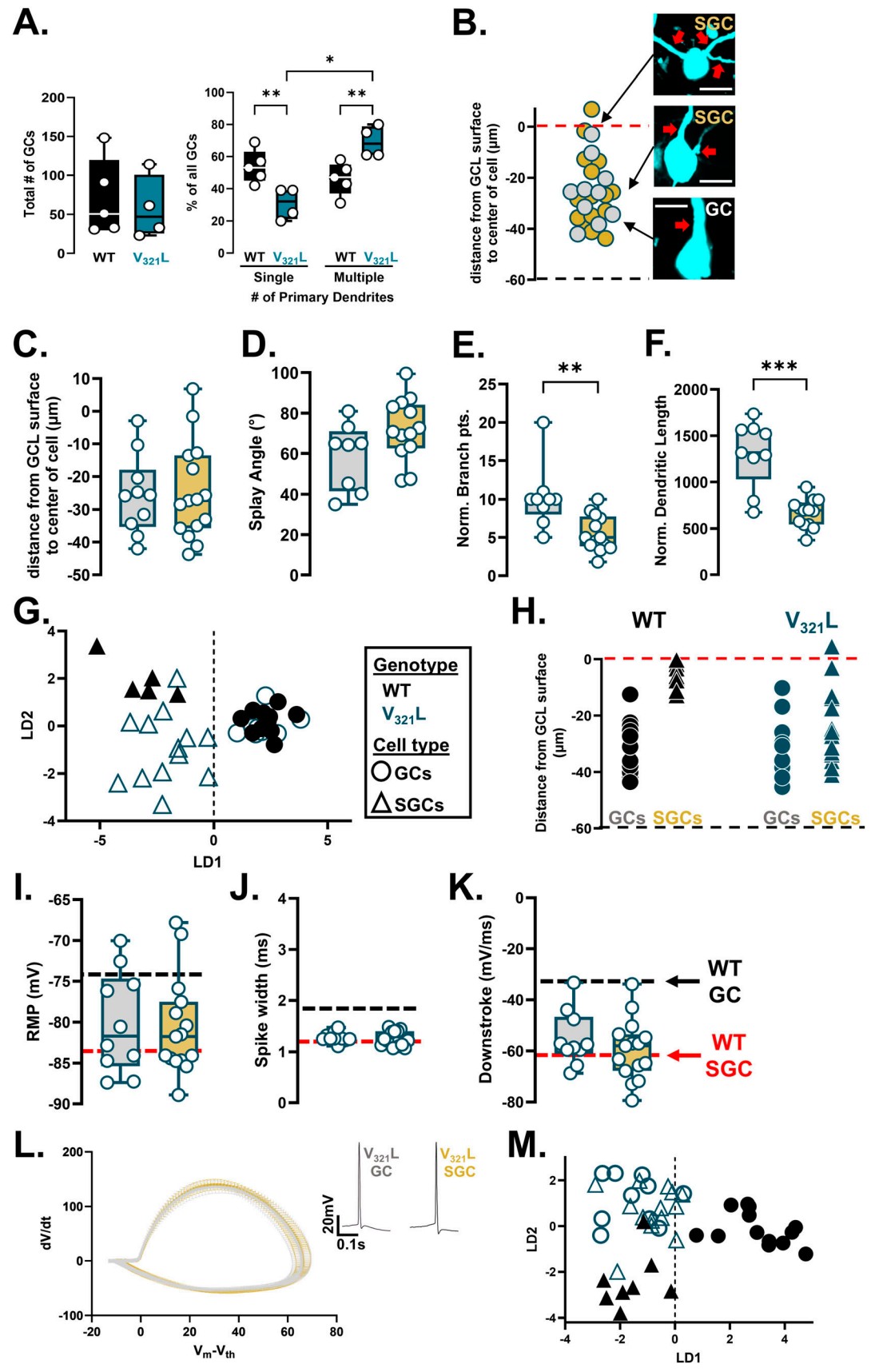

extended this model to $V_{321}L$ mice (see the Materials and Methods section for details).

In WT mice, we found that two electrophysiological properties, spike amplitude ($P$ = 0.042, $\beta1$ = −46.2) and upstroke ($P$ = 0.038, $\beta1$ = −117.88), significantly covaried with cell type when controlling for soma position. However, this model did not detect significant cell-type effects for any electrophysiological properties in $V_{321}L$ mice.

Given the large number of electrophysiological parameters measured (20 parameters) and their potential correlations, we further used a multivariate approach to distinguish cell types based on electrical properties. Principal component analysis (PCA) revealed a clear separation between GCs and SGCs in WT mice along the first two principal components, whereas cells in $V_{321}L$ mice showed substantial overlap (Fig S2E).

To further assess classification accuracy, we trained an LDA classifier with fivefold cross-validation on WT electrophysiological data and applied it to $V_{321}L$ mice. Centroid analysis demonstrated a striking reduction in the separation between GC and SGC centroids in $V_{321}L$ mice along the first linear discriminant (LD1) axis (Fig S2F). The LDA classifier, trained on 80% of the WT data, achieved 78% accuracy in cell-type classification of the remainder 20% WT cells (Cohen's $\kappa$ = 0.57) (Fig S2G, top). Using this classifier on the recordings obtained from $V_{321}L$ mice resulted in 52% classification accuracy (Cohen's $\kappa$ = 0.08) (Fig S2G, bottom). Thus, these data further support the finding of a loss of electrophysiological distinction between GCs and SGCs in the $V_{321}L$ mutant DG.

## $V_{321}L$ mice have a higher number and altered positioning of *Penk+* GCs compared with WT mice

Recent studies have shown that GCs expressing the gene pro-enkephalin (*Penk*) have multiple primary dendrites and a localization consistent with those of embryonically born GCs (Erwin et al, 2020; Mortessagne et al, 2024). Using FISH, we detected *Penk* transcripts in the DG of young adult mice and quantified the distance of *Penk+* cells in the suprapyramidal GCL from the GCL-MOL boundary (Fig 3A). In WT mice, most of the *Penk+* GCs were in the top 2 cell body layers of the suprapyramidal GCL, whereas this bias toward the GCL-MOL boundary was lost in the $V_{321}L$ DG (Fig 3B; $P$ = 0.013).

To examine molecular distinctions between GC subtypes, we performed single-nucleus RNA and ATAC sequencing of single-nucleus suspensions prepared from WT and $V_{321}L$ microdissected DGs (Fig 3C, top). We detected all major cell types across both genotypes indicating no overt differences in clustering or cell composition of the DG between WT and $V_{321}L$ mice (Fig 3C, bottom; Figs S3A and S4A–O; cluster-wise differential gene expression, Table S1). We found no differences in the total number of *Prox1+* neurons (GCs) (Fig S3B; Mann–Whitney test, $P$ = 0.69, U = 6). Upon examination of *Penk* expression in the GC clusters, we found that *Penk*-expressing cells subclustered within the larger clusters of GCs and that the total percentage of *Penk+* GCs was significantly higher in $V_{321}L$ mice (Fig 3D; Mann–Whitney test, $P$ = 0.03, U = 0). We found four clusters of GCs—clusters 0, 1, 5, and 18. Although a greater absolute number of *Penk+* GCs were found in cluster 1 in proximity to cluster 18, cluster 18 contained the highest percentage of *Penk+* cells indicating enrichment for SGCs in cluster 18 in both genotypes (Fig S3C). Marker gene discovery for the different GC clusters revealed relatively minor distinctions in gene expression between clusters 0, 1, and 5. However, cluster 18 was uniquely marked by the expression of Sortilin Related VPS10 Domain Containing Receptor 3 (*Sorcs3*) (Fig S3D, left). We performed FISH for *Sorcs3* transcripts in the DG and found a spatial bias of *Sorcs3+* cells toward the GCL-MOL boundary like *Penk+* cells (and SGCs) consistent with their embryonic origins (Fig S3D, right).

To further assess the specificity of *Sorcs3* as an SGC marker in WT and $V_{321}L$ mice, we quantified the proportions of (*Penk+Sorcs3+*), (*Penk+Sorcs3-*), and (*Penk-Sorcs3+*) GCs in both WT and $V_{321}L$ mutant mice. We found that in line with the increase in the proportion of *Penk+* cells (Fig 3D), $V_{321}L$ mice also showed an increase in the percentage of *Penk+Sorcs3+* GCs (Fig S5A; $P$ = 0.009). We found that there were no significant differences in the proportion of Penk-Sorcs3+ cells between WT and $V_{321}L$ mice (Fig S5A; $P$ = 0.29),

**Figure 2.  GCs in the $V_{321}L$ DG show SGC-like electrical properties.**
**(A)** (Left) Quantification of the total number of GCs from Golgi-stained DG, quantified from WT (black) and $V_{321}L$ (teal) DG. N = 23–148 neurons/mouse from five mice. Each dot is the total number of neurons from one mouse (Welch's *t* test [two-tailed], $P$ = 0.7; t = 0.4338, df = 6.972). (Right) Quantification of the percentage of GCs with single or multiple primary dendrites in WT and $V_{321}L$ DG. N = 23–148 neurons/mouse from 5 mice (two-way ANOVA, Genotype x #Primary dendrites, $P$ = 0.011, F [1,7] = 11.81; $V_{321}L$ single versus $V_{321}L$ multiple, *$P$ = 0.013; WT multiple versus $V_{321}L$ multiple, **$P$ = 0.008; WT single versus $V_{321}L$ single, **$P$ = 0.008). Note: The Golgi staining method is "nonrandom" and biases for labeling of superficially located GCs resulting in disproportionate labeling of SGCs. **(B)** Quantified cell positions of all recorded GCs from $V_{321}L$ mice. The red and black dashed lines correspond to the dorsal and ventral borders of the GCL, respectively. Both SGCs (gold circles) and GCs (silver circles) were found throughout the GCL. Representative images of filled and recorded SGCs and GC are shown. Red arrows point to primary dendrites extending from the cell body. N = 10 cells (GC) and 15 cells (SGC) from 5 mice (17 slices). Scale bar = 10 $\mu m$. **(C)** Quantification of cell body position within the GCL of GCs (silver) and SGCs (gold). SGC cell body locations are not significantly different to those of GCs ($P$ = 0.9012, Welch's *t* test [two-tailed]; t = 0.1256, df = 21.78; N = 10 cells [GC] and 15 cells [SGC] from 5 mice). **(D)** Quantification of splay angle of GCs and SGCs. The splay angle was not significantly different between GCs and SGCs ($P$ = 0.0811, Welch's *t* test [two-tailed]; t = 1.879, df = 14.11; N = 8 cells [GC] and 13 cells [SGC] from 5 mice). **(E)** Number of normalized branch points was significantly higher in GCs compared with SGCs (**$P$ = 0.0013, Mann–Whitney test, U = 13; N = 9 cells [GC] and 13 cells [SGC] from 5 mice). **(F)** Normalized dendritic length was significantly higher in GCs compared with SGCs (***$P$ = 0.0005, Welch's *t* test [two-tailed]; t = 5.023, df = 10.05; N = 9 cells [GC] and 13 cells [SGC] from 5 mice). **(G)** Linear discriminant analysis of morphological properties of WT (black) and $V_{321}L$ (teal) cells reveals separation of GCs (circles) and SGCs (triangles) along the LD1 axis. **(H)** $V_{321}L$ SGCs are localized throughout the GCL unlike WT SGCs. Each symbol represents a single cell from biocytin-filled reconstructions replotted from Panel (B) and Fig 1C—WT (black) and V321L (teal); GCs (circles) and SGCs (triangles). A red dashed line represents the dorsal edge of the GCL. **(I, J, K)** $V_{321}L$ GCs and SGCs did not show significant differences in their electrophysiological properties. **(I)** RMP ($P$ = 0.8374, Welch's *t* test; t = 0.2081, df = 18.86). **(J)** Spike width ($P$ = 0.2776, Mann–Whitney test, U = 55). **(K)** Downstroke ($P$ = 0.5392, Mann–Whitney test, U = 63.5). A black dashed line indicates the median values for WT GCs, and a red dashed line indicates median values for WT SGCs. N = 10 cells (GC) and 15 cells (SGC) from 5 mice. **(L)** Phase plots of action potential dynamics (dV/dt versus Vm-Vth) provide a visualization of the AP waveform of GCs and SGCs. N = 10 cells (GCs) and 15 cells (SGCs) from 5 mice. Inset shows representative action potential waveforms of a GC and SGC recorded at rheobase. **(M)** Linear discriminant analysis of electrical properties of WT (black) and $V_{321}L$ (teal) cells reveals separation of WT GCs (circles) from all other cells from both genotypes along the LD1 axis.

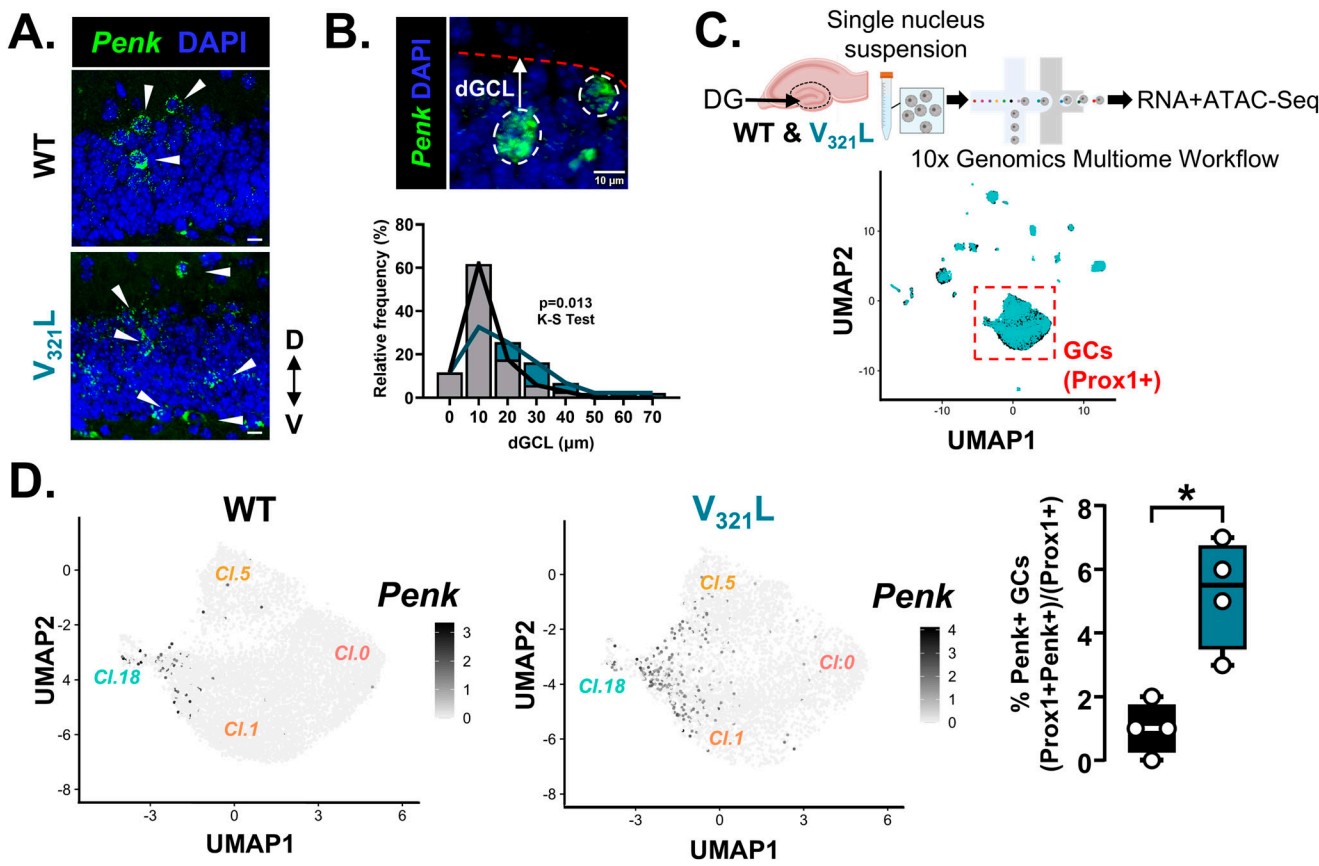

**Figure 3. V₃₂₁L mice have a higher number and altered positioning of *Penk*+ GCs compared with WT mice.**
**(A)** (Top) Representative image of *Penk* mRNA (green) expression in the DG of a WT mouse. (Bottom) Representative image of *Penk* mRNA (green) expression in the DG of a V₃₂₁L mouse. White arrowheads indicate *Penk*+ cells in the GCL and IML. Anatomical guide indicates dorsal (D) and ventral (V) directions. **(B)** (Top) Representative image showing two *Penk*+ GCs outlined with white dashed lines. The white arrow denotes the distance of the *Penk*+ cell from the GCL-MOL boundary (dGCL). The red dotted line delineates the GCL-MOL boundary of the suprapyramidal blade (SPB) of the DG. Scale bar = 10 μm. (Bottom) Quantification of the soma positions within the GCL of the *Penk*+ cells from WT (black) and V₃₂₁L (teal) mice. The distribution of *Penk*+ cells in the V₃₂₁L DG was significantly altered compared with WT DG (Kolmogorov–Smirnov test, *P* = 0.013, D = 0.36). N = 3 mice/genotype. **(C)** (Top) Schematic of the workflow for single-nucleus RNA- and ATAC-Seq (Multiome) of WT and V₃₂₁L DG. (Bottom) UMAP of snRNA-Seq showing clusters of WT (black) and V₃₂₁L (teal) nuclei. The red dashed box denotes clusters of GCs identified as *Prox1*+ excitatory neurons (see Fig S3A and B). N = 4 mice/genotype, two male and two female. **(D)** UMAPs showing the expression of *Penk* in GCs of WT (left) and V₃₂₁L (middle) mice. The level of expression is indicated by the heatmap alongside. (Right) Quantification of the percentage of GCs expressing *Penk*+ in WT and V₃₂₁L mice showing that V₃₂₁L mice had significantly more *Penk*+ GCs (Mann–Whitney test [two-tailed], *P* = 0.03, U = 0). N = 4 mice/genotype.

indicating that the mutation does not affect *Sorcs3* expression outside the *Penk*+ population of GCs indicating its retained specificity in the mutant mice. However, upon examining the proportions of *Penk*+*Sorcs3*- GCs, we found that this population of GCs was also expanded in the V₃₂₁L mutant mice (Fig S5A; *P* = 0.0002) indicating the potential presence of further heterogeneity within the *Penk*+ (SGC) population. Upon examination of clustering of these subpopulations, we found that *Penk*+*Sorcs3*- cells were excluded from cluster 18 in both WT and V₃₂₁L mice (Fig S5B and C).

### Discovery of novel marker genes for SGCs

We observed greater numbers and altered localization of *Penk*+ GCs in the V₃₂₁L DG (Fig 3A, B, and D). We identified *Sorcs3* as a marker for GCs located in the *Penk*-enriched GC cluster (cl.18) (Fig S3D). Thus, we next asked if V₃₂₁L mice had greater numbers and/or altered localization of *Penk*- and *Sorcs3*-co-expressing GCs using FISH. We

found that V₃₂₁L mice had significantly higher numbers of cells co-expressing *Penk* and *Sorcs3* (Fig 4A, *P* = 0.014). We also found a significantly higher number of *Penk*- and *Sorcs3*-co-expressing cells in V321L mice using snRNA-Seq (Fig 4b", *P* = 0.03, FDR corrected q = 0.002). In addition, we noted that the *Penk*+*Sorcs3*+ cells in the V₃₂₁L DG were not restricted to the GCL-MOL boundary cell layers (Fig 4a', *P* = 0.029), much like the morphologically identified SGCs (Fig 2H).

To ascertain whether the increase in *Penk*+ or *Penk*+*Sorcs3*+ GCs reflected up-regulation of *Penk* and/or *Sorcs3* because of the loss of Nrg1 nuclear back-signaling, or whether it indeed reflected an increase in molecularly defined SGCs, we sought to identify additional genetic markers for this cell type. We reasoned that combinatorial detection of such markers could discern changes in cell-type composition versus changes in regulation of *Penk* expression. We assessed differential gene expression in *Penk*+ GCs compared with *Penk*-negative GCs yielding 102 genes whose expression was

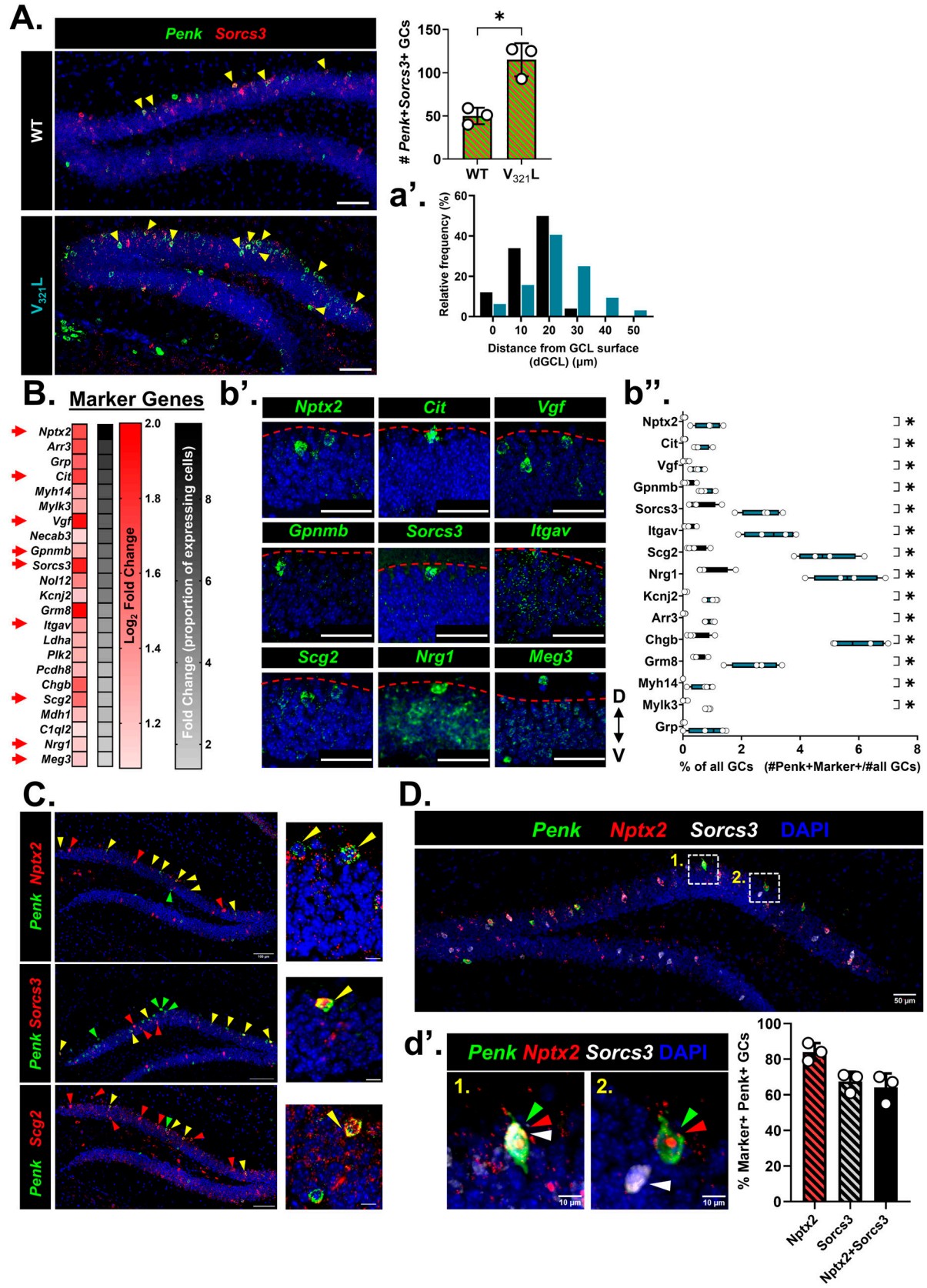

significantly enriched in *Penk+* GCs (fold change of ≥2 and adj. *P* ≤ 0.05). We rank-ordered these genes based on the ratio of percentage of *Penk+* cells expressing them to the percentage of *Penk-* cells expressing them; 23 of these genes are displayed in Fig 4B (see Table S2). Of these, we selected nine genes for further validation using FISH to assess spatial localization consistent with embryonically born GCs. We found that different genes displayed varying degrees of spatial bias toward the GCL-MOL boundary with some showing a more exclusive spatial segregation such as neuronal pentraxin 2 (*Nptx2*), Citron (*Cit*), and *Sorcs3*, which we showed earlier to also be a marker gene for cluster 18—the cluster enriched for *Penk+* GCs (Fig 4b'). We next quantified the percentage of *Penk+* GCs co-expressing each of these marker genes in the WT and V$_{321}$L DG snRNA-Seq datasets. Except for gastrin-releasing peptide (*Grp*) (Mann–Whitney test, *P* = 0.1, two-stage step-up Benjamini–Krieger–Yekutieli FDR corrected for multiple comparisons, q = 0.01, U = 2.5), all other marker gene Penk-co-expressing cells formed a significantly higher proportion of all GCs in the V$_{321}$L mice (Fig 4b", *P* = 0.03, FDR corrected q = 0.002). Intriguingly, this increase in numbers of *marker+Penk+* cells was only observed in *Penk* and *Nrg1* co-expressing GCs (Fig S6A) and not in *Nrg1*-non-expressing *Penk+* GCs (Fig S6B). In FISH experiments for *Penk* and *Nrg1*, we found that most of the *Penk+* GCs co-expressed *Nrg1* (~94%) (Fig S6B, inset). We next performed FISH to detect the co-expression of *Penk* with candidate markers such as *Nptx2*, *Sorcs3*, and *Scg2* and found strong overlap between *Penk* and the selected marker genes, particularly biased toward the GCL-MOL boundary of the suprapyramidal blade of the DG (Fig 4C). We selected *Nptx2* and *Sorcs3* given their higher relative exclusivity for expression toward the GCL-MOL boundary compared with *Scg2*. We quantified the degree of overlap between *Penk* and *Nptx2* or *Sorcs3* and of all three genes using FISH. We found that on average 84% of the *Penk+* GCs also expressed *Nptx2*, and 67% of the *Penk+* GCs also expressed *Sorcs3*. The co-expression of *Nptx2* and *Sorcs3* was detected in 64% of the *Penk+* GCs (Fig 4D and d'). Taken together with the restriction of

*Penk-* and *Sorcs3*-co-expressing cells in the cell body layers at the GCL-MOL boundary (Fig 4a') and the recently reported SGC-like morphology of *Penk+* GCs (Mortessagne et al, 2024), these data indicate that the *Penk*, *Nptx2*, and *Sorcs3* co-expressing population of GCs represents SGCs.

## Non-SGCs harbor the potential to express an SGC-like transcriptomic state

SGCs are thought to be generated exclusively during embryonic development; however, several lines of evidence suggest that the adult DG possesses the capacity to produce SGCs (Lauer et al, 2003; Kim et al, 2009; Fitzsimons et al, 2013; Llorens-Martin et al, 2014; Terreros-Roncal et al, 2019; Mortessagne et al, 2024). We found that the SGC-enriched cluster of GCs (cl.18) was closely related to other GC clusters. Marker gene discovery was unable to identify any gene with unique expression that distinguished clusters 0, 1, and 5 from each other or from cluster 18 (Fig S3D). However, cluster 18 GCs showed the expression of *Sorcs3* as a unique marker, suggesting that the SGC-like transcriptomic state could be conceptualized as a modular "add-on extension" to a general GC-like transcriptome. To test this "add-on" concept, we performed trajectory analysis on the GC clusters using Monocle3 (Cao et al, 2019). To assign the root node, we assessed the expression of *Camk4*, which has been shown to transiently increase in expression in maturing GCs and *Ntng1*, whose expression gradually increases and peaks as GCs mature (Hochgerner et al, 2018) (Fig 5A and a'). FISH for *Camk4* and *Ntng1* revealed that they were expressed in a spatial gradient in the DG GCL such that *Camk4*-expressing cells occupied the lower half of the GCL in proximity to the hilus (Hi), whereas the *Ntng1*-expressing cells occupied the upper half in proximity to the molecular layer (MOL) (Fig 5A). Because the DG laminates "outside-in," *Camk4+* GCs represent GCs with a more recent birthdate compared with *Ntng1+* GCs. We examined the expression of *Camk4* and *Ntng1* in our GC clusters and found opposing gradients of expression in the UMAPs

**Figure 4. Discovery of novel marker genes for semilunar granule cells.**
**(A)** Co-expression of *Penk* and *Sorcs3* was evaluated in the V$_{321}$L DG. Representative image from a 1-mo-old WT and V$_{321}$L DG showing cells expressing *Penk* (green) and *Sorcs3* (red). Yellow arrowheads indicate *Penk+Sorcs3+* cells. Scale bar = 100 µm. (Right) Quantification of the number of *Penk+Sorcs3+* cells in the suprapyramidal blade (SPB) of the DG of WT and V$_{321}$L mice. N = 9 sections from three mice; each dot represents one mouse. There were significantly more *Penk+Sorcs3+* cells in the V$_{321}$L DG (Welch's *t* test (two-tailed), *P* = 0.014; t = 5.278, df = 2.941). (a') Frequency distribution of location (dGCL) of *Penk+Sorcs3+* cells in the SPB of the DG of WT and V$_{321}$L mice (N = 3 mice each). The spatial distribution of labeled cells was significantly different between the genotypes, with the V$_{321}$L mice showing more Penk+Sorcs3+ cells present in the lower layers of the GCL, away from the GCL-MOL boundary (K-S test, *P* = 0.029, D = 0.4). **(B)** Marker gene discovery for *Prox1+Penk+* GCs (SGCs) identified several genes whose expression was enriched in SGCs. The red gradient heatmap shows the expression of the designated genes as log2 fold change relative to non-SGCs (*Prox1+Penk-*GCs). The accompanying black gradient heatmap indicates the degree of exclusivity of the expression of the identified putative SGC marker genes (% *Penk+* GCs/% *Penk-*GCs). Red arrows indicate genes chosen for validation via ISH. (b') Expression of the indicated subset of the putative marker genes was evaluated in the young adult DG (2 mo) using RNAScope ISH. Images of the middle of the GCL in the suprapyramidal blade of the DG were examined. Most of the genes enriched in *Penk+* GCs were found to be highly expressed in cells located closer to/at or beyond the boundary between the GCL and MOL (red dashed line) in line with stereotypical location of SGCs. Scale bar = 50 µm. (b") Quantification of % of *Penk+* GCs expressing selected marker genes in WT and V$_{321}$L mice. A number of cells co-expressing *Penk* and each of the marker genes were obtained from the snRNA-Seq data and divided by the total number of cells in clusters 0, 1, 5, and 18. WT and V$_{321}$L samples were compared for each gene using multiple Mann–Whitney tests with an FDR correction. With the exception of Grp (WT versus V321L, FDR adj.*P* = 0.14, U = 2.5), V321L samples had significantly more cells co-expressing Penk and selected marker genes (WT versus V321L, FDR adj.*P* = 0.03, U = 0) (N = 4 mice/genotype; 2M, 2F). **(C)** Co-expression of *Penk* (green) and several marker genes (red) was evaluated using RNAScope multiplex ISH. Images showing co-expression of *Penk* with either *Nptx2* (top), *Sorcs3* (middle), or *Scg2* (bottom) are shown. Scale bar = 100 µm. Red arrowheads indicate cells expressing marker gene alone, green arrows indicate cells expressing *Penk* alone, and yellow arrows indicate cells co-expressing *Penk* and marker gene. Inset shows a higher magnification image of a few example cells with *Penk*-marker co-expression (yellow arrowheads). Scale bar = 10 µm. **(D)** Co-expression of *Penk*, *Nptx2*, and *Sorcs3* was evaluated as a defining feature for identification of SGCs. Representative image from a 2-mo-old WT DG showing cells expressing *Penk* (green), *Nptx2* (red), and *Sorcs3* (white). Scale bar = 50 µm. Insets labeled 1 and 2 show cells expressing different combinations of the selected markers. Inset 1 shows a *Penk+Nptx2+Sorcs3+* cell. (d') (Left) Inset 2 shows two labeled cells—one expressing *Sorcs3* alone (white arrowhead) and another expressing *Penk* and *Nptx2* (green and red arrowheads). Scale bar = 10 µm. (Right) Quantification of the percentage of *Penk+* cells expressing the selected marker gene combinations in the suprapyramidal blade of the DG of WT mice. N = 9 sections from three mice; each dot represents one mouse. Percentage of *Penk+marker+* cells is as follows (mean ± std.dev): *Nptx2* (red with black patterned fill), 84 ± 5%, *Sorcs3* (gray with black patterned fill), 67.3 ± 5.5%, and *Nptx2+Sorcs3* (black filled bar), 64 ± 7.9%.

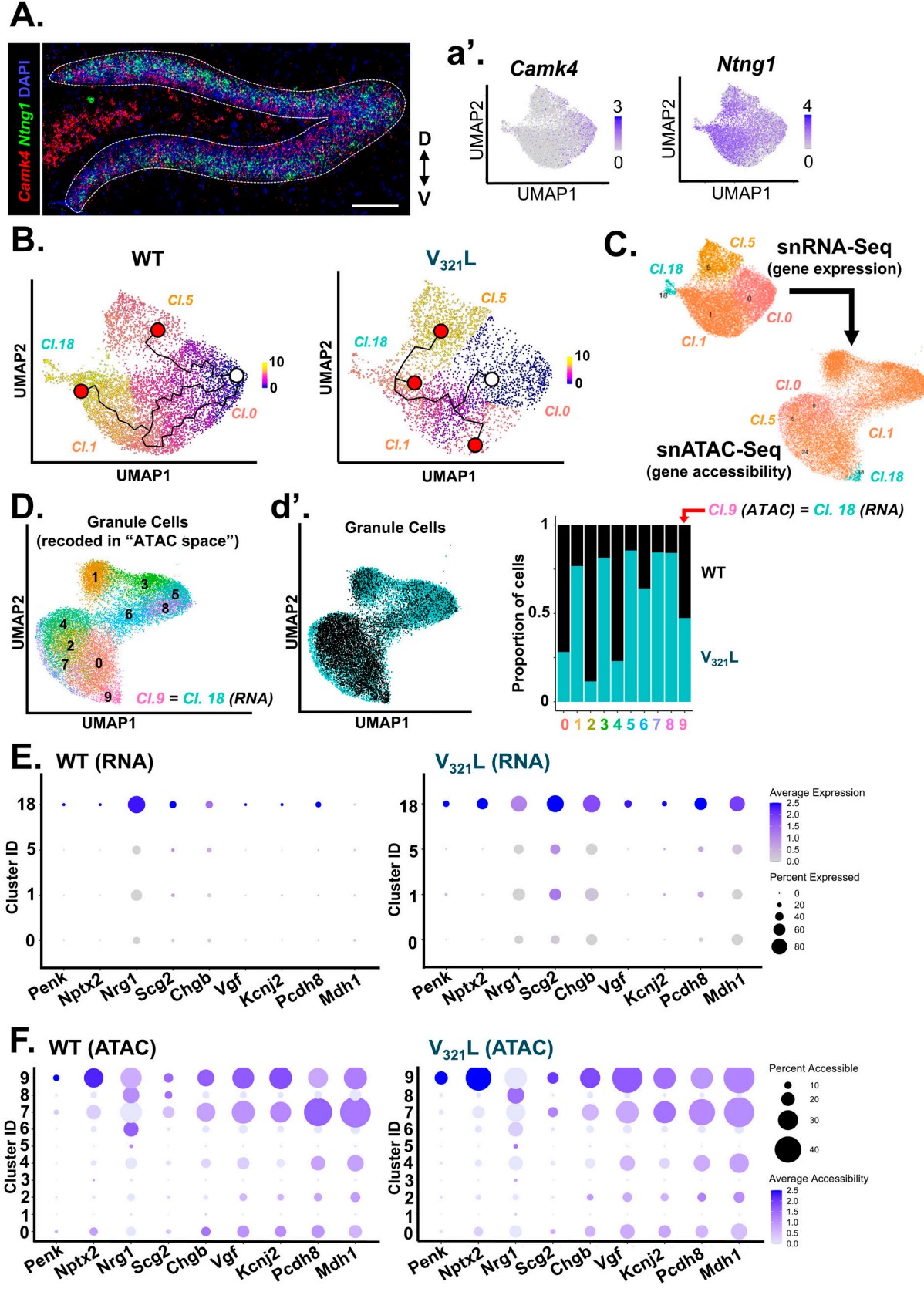

such that cluster 0 GCs represented the younger GCs occupying the bottom half of the GCL, whereas GCs located along the continuum from cluster 0 to cluster 18 showed gradually increasing *Ntng1* expression, likely representing GCs located in the top half of the GCL (Fig 5a'). Next, we computed pseudotime for the cells falling along these trajectories (Fig 5B). The top gene explaining pseudotime in WT DG was found to be *Sorcs3*, the cluster 18 marker gene that we also found to be a marker gene for SGCs (Table S3). In line with this, cells with late pseudotime were found toward the end of cluster 1 and in cluster 18, where we previously noted the presence of *Penk+* GCs (Fig 3D). Trajectories in the $V_{321}L$ DG were highly altered, and cells with late pseudotime were present in cluster 5 (Fig 5B, right). The cluster 1–18 transition area in $V_{321}L$ clusters fell along an intermediate pseudotime; however, *Sorcs3* expression still faithfully represented the cells in this zone indicating the preserved molecular profile of SGCs in the $V_{321}L$ DG. To address differences in cell sampling between WT and $V_{321}L$ GCs, we performed downsampling of WT GCs and recalculated pseudotime and gene expression associated with pseudotime (Fig S7A). We found that the pseudotime ordering of GCs and associated gene expression states were robust to differences in sampling frequency, with *Sorcs3* still representing cells with the maximal pseudotime (Fig S7B). We next used "Trajectory Geometry" to assess whether the identified trajectories in WT and $V_{321}L$ mice have well-defined directionality (Laddach et al, 2024). First, we tested cells along the identified trajectories in the WT mice (original and downsampled) against 1,000 iterations of pseudotime randomized cell trajectories. WT original and downsampled trajectories resulted in significantly lower spherical distances compared with randomized ones indicating the presence of a definite directionality within the trajectories (Wilcoxon's signed rank test, randomized versus WT original, $P = 4.25 \times 10^{-11}$, randomized versus WT downsampled, $P = 8.00 \times 10^{-31}$). Similarly, the $V_{321}L$ trajectory also produced a significantly smaller spherical distance relative to randomized trajectories (Wilcoxon's signed rank test, randomized versus $V_{321}L$, $P = 3.17 \times 10^{-54}$). Interestingly, the spherical distances obtained for the WT trajectories were greater than those for the $V_{321}L$ trajectory indicating the presence of more complex fate decisions as seen by the branching in the WT trajectory. Importantly, although both the original and downsampled

WT data yielded valid and statistically significant trajectories, they reflect different levels of complexity based on sampling. Thus, trajectory inference is not exhaustive but rather an approximation of lineage dynamics based on available cellular content.

Taken together, these data indicate that SGCs can be placed along a molecular continuum of GC maturation and that the SGC transcriptomic phenotype is largely unaltered in the $V_{321}L$ DG. Acquisition of an SGC-like transcriptomic state would necessitate accessibility of SGC-enriched genes along with the presence of the appropriate transcriptional regulators, which can activate/derepress the SGC-enriched genes. To investigate the former of these two requirements, we examined chromatin accessibility in the same GCs that were profiled for gene expression. We remapped the GCs identified based on their gene expression profiles into the "ATAC space" (Fig 5C). We noticed that GCs from clusters 0 and 18 preserved their within-cluster cohesion continuing to cocluster in a single ATAC cluster each. However, GCs from clusters 1 and 5 were found to distribute into eight smaller subclusters (Fig 5D). These results potentially indicate a more dynamic restructuring of the chromatin landscape in GCs that represented the intermediate pseudotimes. We also noted that there was a dramatic difference in the distribution and clustering of GCs from WT and $V_{321}L$ DG based on chromatin accessibility (Fig 5d', left). Cluster 9 was comprised of the *Penk+* GCs, and it was uniformly populated by WT and $V_{321}L$ GCs; however, cluster 4 that was comprised of the *Camk4+* GCs (cluster 0 in the "RNA space") was disproportionately populated by WT GCs, which made up ~75% of all cells in that cluster. These results indicate that $V_{321}L$ GCs might escape a "younger" chromatin state. To investigate this further, we performed trajectory analysis and pseudotime mapping of the ATAC-Seq data (Fig S8). Once again, we found that the *Penk+* GCs (now located in ATAC cluster 9) had the maximum pseudotime values based on chromatin accessibility (Fig S8A, left). Intriguingly, in the analysis of WT GCs, there were three end points to the trajectories, one of which terminated at the cluster with the maximal pseudotime. However, in the $V_{321}L$ GCs, there was only one end point terminating at cluster 9 (Fig S8A, right). We evaluated the accessibility of the *Penk* transcription start site (TSS) (±1 kb) and found that accessibility of the *Penk* TSS increased along pseudotime in both WT and $V_{321}L$ GCs, but the trajectories

**Figure 5. Non-SGCs harbor the potential to express an SGC-like transcriptomic state.**
**(A)** Representative image of a WT DG showing the expression of *Camk4* (red) and *Ntng1* (green). The DG blades are outlined with a white dashed line. Scale bar = 100 μm. The anatomical legend indicates orientation of the slice (D = dorsal, V = ventral). **(A)** (a') UMAPs showing the expression of the early-stage maturation marker gene *Camk4* (left) and late-stage maturation marker *Ntng1* (right) with opposing gradients of expression within the granule cell clusters similar to their opposing gradients of expression within the GCL as shown in Panel (A). **(B)** Trajectory mapping using Monocle3 reveals different trajectories between WT and $V_{321}L$ GCs. Pseudotemporal ordering of cells was performed by assigning root nodes in cluster 0 owing to higher *Camk4* expression. (Left) In WT GCs, cluster 18 and adjoining cells in cluster 1 have the highest pseudotime, whereas cluster 5, which hosts one of the trajectory end points, is at an intermediate pseudotime. (Right) In $V_{321}L$ GCs, cluster 18 and adjoining cells in cluster 1 have an intermediate pseudotime with a branched trajectory terminating at that junction. Meanwhile, cluster 5 houses cells with the highest pseudotime. The white circle indicates the root node for the trajectories, and the red circles indicate end points for the different trajectories. **(C)** GCs from snRNA-Seq clusters (clusters 0,1,5, and 18) were assessed for their chromatin accessibility using ATAC-Seq. UMAPs show GC clusters based on their gene expression profiles (RNA-Seq, top) and the same GCs clustered based on their gene accessibility profiles (ATAC-Seq, bottom). Colors and cluster IDs indicate identity based on gene expression. **(D)** GCs were clustered based on gene accessibility (ATAC-Seq) and assigned new cluster IDs. Note that RNA-Seq cluster 18 GCs maintained their own cluster in the "ATAC space" as cluster 9. (d') (Left) UMAP of snATAC-Seq showing clusters of WT (black) and $V_{321}L$ (teal) nuclei. (Right) Quantification of proportional makeup of each of the snATAC-Seq clusters based on the genotype (WT versus $V_{321}L$) of the nuclei. Except for cluster 9, all the other clusters were either biased toward WT or $V_{321}L$ nuclei. **(E)** Dot plot for nine example SGC-enriched genes showing average gene expression (snRNA-Seq) represented by the color of the dots. Percentage of cells with the expression of the specified gene within a given cluster is represented by the size of the dots according to the accompanying legend. Data from GCs—clusters 0, 1, 5, and 18—are shown split by genotype—WT (left) and $V_{321}L$ (right). Note that the expression of SGC-enriched genes is largely restricted to cluster 18 in WT GCs. **(F)** Dot plot for nine example SGC-enriched genes showing average gene accessibility (snATAC-Seq) represented by the color of the dots. Percentage of cells with accessibility of the specified gene within a given cluster is represented by the size of the dots according to the accompanying legend. Data from GCs—clusters 0–9—are shown split by genotype—WT (left) and $V_{321}L$ (right). Note, cluster 18 GCs from Panel E are represented in cluster 9. Thus, genes enriched for expression in SGCs are broadly accessible to GCs.

traversed different clusters (Fig S8B and b'). Surprisingly, unlike the WT-V$_{321}$L difference in total number of cells expressing *Penk* (Fig 1F), the total number of GCs having the *Penk* locus accessible was not significantly different between the genotypes (Fig S8b"; *P* = 0.9). Therefore, next we compared the expression and accessibility of the SGC-enriched genes in WT and V$_{321}$L GCs. We examined the expression of a select set of SGC-enriched genes for which there were RNA and ATAC counts present for each cell (Fig 5E and F). As shown earlier, there were more cells expressing SGC-enriched genes in V$_{321}$L mice and the expression was not as restricted to cluster 18 compared with the WT GCs (Fig 5E). Cluster 18 GCs (RNA-Seq) were remapped to cluster 9 (ATAC-Seq), and we found accessibility for each of the marker genes was present in cluster 9 of both WT and V$_{321}$L GCs (Fig 5F). However, we also found SGC-enriched genes to be accessible in cells in nearly all other clusters for both genotypes.

These data indicate that GCs in WT mice might use an active repressive mechanism to prevent the expression of SGC marker genes in non-SGCs, and this mechanism might be disrupted in the V$_{321}$L DG. To investigate this further, we searched for putative regulators for the SGC-enriched genes using ChEA and ENCODE databases. We identified 60 genes whose expression was enriched in *Penk+* GCs and which were also correlated with maximal pseudotime in the trajectory analysis of GC gene expression (Fig 5B; Table S1). We found strong enrichment for regulation of these genes by Suz12, a core component of the polycomb repressive complex 2 (PRC2) (Table S4).

Thus, taken together these data indicate that an SGC-like transcriptomic state is accessible to non-SGCs, but is kept repressed by Nrg1 nuclear back-signaling potentially through regulation of the PRC2 (Rajebhosale et al, 2024). Loss of Nrg1 nuclear back-signaling because of V$_{321}$L mutation might result in derepression of the SGC genes and subsequent acquisition of an SGC-like transcriptome by non-SGCs.

## SGC numbers in the DG increase during the adolescence–adulthood transition in WT mice

Our data predicted that repression of SGC genes is mediated at least in part by PRC2. The role of PRC2 in lineage specification is well studied; recent reports have shown that PRC2 dictates the timeline of neuronal maturation (Ciceri et al, 2024). In line with this, a recent study showed age-dependent increase in *Penk* expression in the DG GCL, which was not associated with adult-born or immature GCs (Mortessagne et al, 2024). Thus, using the co-expression of *Nptx2*, *Sorcs3*, and *Penk* as an SGC "identifier" we quantified SGCs in the mouse DG of 1-, 2-, 3-, and 12-mo-old WT mice (Fig 6A and B). We confirmed that WT mice indeed show an increase in *Penk* expression starting between 2 and 3 mo of age, a period marking puberty and transition to early adulthood (Fig 6a'; *P* = 0.0032, post hoc multiple comparisons: 1 versus 3 mo, *P* = 0.008, 1 versus 12 mo, *P* = 0.03, 2 versus 3 mo, *P* = 0.008). We found that the numbers of *Penk+Nptx2+Sorcs3+* GCs also increased with age (Fig 6b'; *P* = 0.0012, 1 versus 12 mo, Dunn's corrected *P* = 0.039). These dynamics of SGC abundance over adolescence were lost in the V$_{321}$L DG (Fig S9A). There were no significant differences in the overall percentage of *Nptx2-* or *Sorcs3-*expressing *Penk+* GCs at any of the ages, indicating that this combination of markers stably reports this population of

GCs over aging in both genotypes (Fig S9B). Because *Penk+* GCs are embryonically generated and bulk of the GCs are not, we wondered whether the SGC-like transcriptome might simply reflect cellular age and not necessarily mark SGCs. We examined ISH data from embryonic day 18.5 (E18.5) mouse DG (Allen Brain Atlas), when the newborn SGCs would be between 0 and 3 d of age, for the expression of a subset of the SGC-enriched genes from our snRNA-Seq analysis (Fig S9C). We found cells expressing SGC-enriched genes such as *Penk*, *Scg2*, *Nrg1*, and *Cit* in the dorsal border of the developing GCL at E18.5, indicating that marker gene expression is unlikely to be an artifact of cellular age (Fig S9C).

In addition, we did not find any *Penk+* cells that expressed the immature neuron marker doublecortin (*Dcx*) at 3 mo of age, indicating that the source of the increase in SGC marker+ cells during the adolescence-to-adulthood transition is not likely to be immature GCs in line with predictions from the trajectory analyses, which posit a gain of an SGC phenotype in mature GCs (Fig S9D). Thus, we sought to identify genes whose expression delineates the cell-type transitions toward SGC-enriched cluster 18 (Fig S10A). Using Monocle3, we identified modules of genes whose expression most strongly correlated with end-trajectories toward cluster 18 resulting in the identification of four significantly correlated modules (Fig S10B). We identified genes within these end-trajectory modules that overlapped with our previously identified SGC-enriched gene set with the rationale that genes whose expression delineates transitions toward cluster 18, and maintained higher expression in Penk+ GCs are more likely to represent key components of the SGC phenotype; we identified 60 genes that met this criterion (Fig S10C). Subjecting these genes to ontology analyses for cellular and biological processes, we identified them to be enriched for the regulation of a variety of neuronal functions with the feature of highest odds ratio being regulation of RMP, in line with our results showing a more hyperpolarized RMP in morphologically identified SGCs (Fig 1J). This enrichment was driven by two potassium channel genes—*Kcnk1* and *Kcnj2* (Fig S10D). Examination of the expression of these genes showed that both had a bias for expression along the pseudotime continuum shown in Fig 5, with more consistent and higher expression toward cluster 18 (Fig S10E). Like our findings regarding SGCs, we noted the higher and widespread expression of both *Kcnk1* and *Kcnj2* in the V$_{321}$L GCs (Fig S10F). Thus, the identified 60 genes are good candidates for assessment of unique functional aspects of SGCs. These 60 genes were also enriched for significant associations with a variety of neuropsychiatric phenotypes and traits such as sensory sensitization, psychosis, schizophrenia, bipolar disorder, and neuroticism (Table S5).

## Natural decline in capacity for Nrg1 nuclear back-signaling corresponds to the SGC fate

SGCs express high levels of *Nrg1* implying a role of Nrg1 signaling in SGC function (Fig 4B). We also found that a loss of Nrg1 nuclear back-signaling because of the V$_{321}$L mutation resulted in more SGCs, implying that intact Nrg1 nuclear signaling suppresses the SGC fate (Figs 2, 3, and 4). To reconcile these findings, we postulated that Nrg1 expressed in WT SGCs might have a reduced capacity to engage in nuclear signaling. Nuclear back-signaling by Nrg1 has been shown

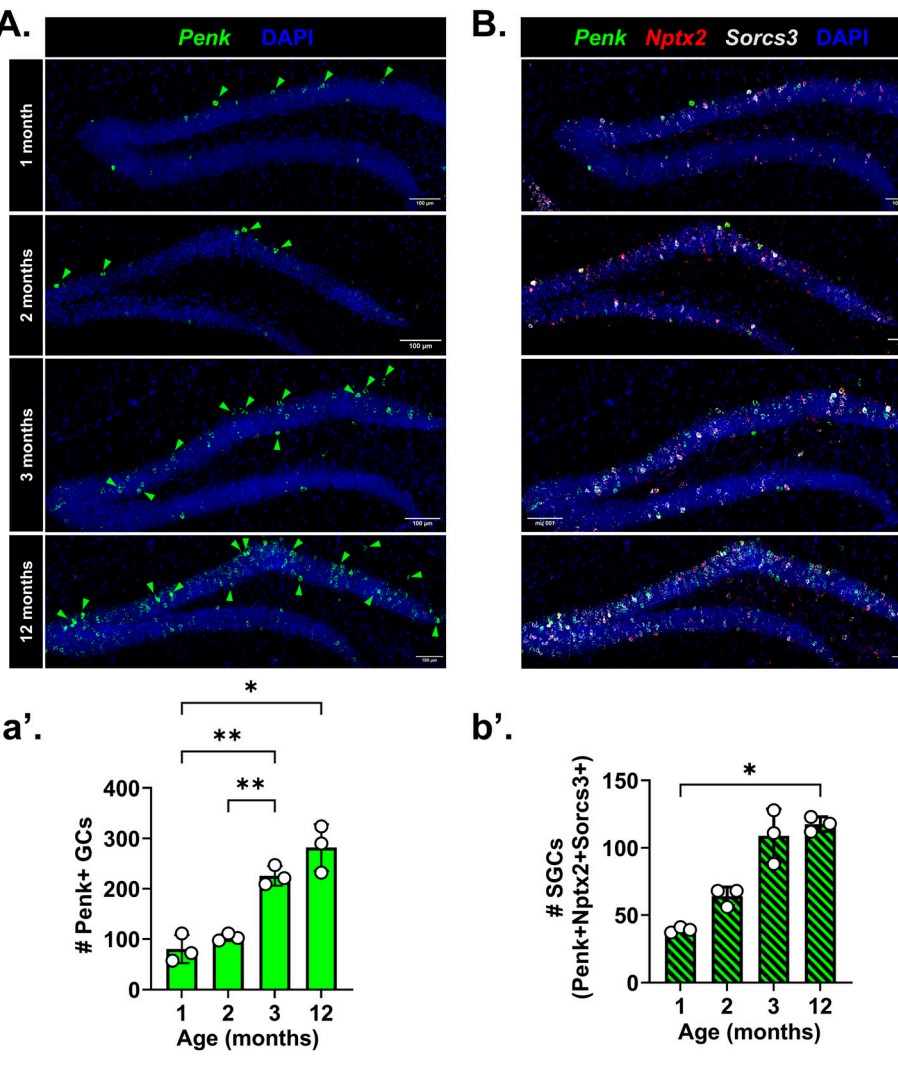

**Figure 6. Total number of SGCs increases over the adolescence-to-adulthood transition in WT mice.**
**(A)** Representative images of WT DG showing the expression of *Penk* (green) using ISH at different ages: 1, 2, 3, and 12 mo[$]. Green arrowheads indicate *Penk*+ cells in the GCL. Scale bar = 100 μm. (a') Quantification of the total number of *Penk*+ cells in the GCL of the suprapyramidal blade over age (Brown–Forsythe ANOVA, P = 0.002, F = 33.4[3, 4.2]; Dunnett's T3 multiple comparisons test: 1 versus 3 mo, **P = 0.008, 1 versus 12 mo, *P = 0.03, 2 versus 3 mo, **P = 0.008). [$]Note that the 12-mo-old group comprised of animals aged 9–12 mo.
**(B)** Representative images of WT DG showing the expression of *Penk* (green), *Nptx2* (red), and *Sorcs3* (white) using ISH at different ages: 1, 2, 3, 12 mo[$]. Scale bar = 100 μm. (b') Quantification of the total number of *Penk+Nptx2+Sorcs3+* cells in the GCL of the suprapyramidal blade over age (Kruskal–Wallis test, P = 0.001, KW = 9.5; Dunn's multiple comparisons test: 1 versus 12 mo, *P = 0.039). [$]Note that the 12-mo-old group comprised of animals aged 9–12 mo.

to be carried out by the Type III isoform of Nrg1 (Bao et al, 2003). Thus, we examined the expression of *Type III Nrg1*, in the DG over age hypothesizing that a reduction in the expression of *Type III Nrg1* might correspond to the adolescence-to-adulthood transition period when the SGC marker+ cells increase. We found that *Type III Nrg1* expression declined from 1 to 12 mo of age in WT mice; however, the reduction between 1 and 3 mo of age was not significant (Fig 7A; Kruskal–Wallis test, P = 0.01, KW = 6.25, 1 versus 12 mo, Dunn's corrected P = 0.04).

*Bace1* is a critical regulator of nuclear back-signaling. Cleavage by *Bace1* is necessary for producing Nrg1 proteins capable of being processed by gamma secretase (Willem et al, 2006; Vullhorst et al, 2017). Thus, we examined the expression of *Bace1* along pseudotime trajectories. Bace1 expression was found to be higher in cells at intermediate pseudotime compared with the beginning or end of the trajectories, indicating low *Bace1* expression in SGCs (Fig 7B). Thus, although SGCs have higher expression of *Nrg1* relative to non-SGCs (Fig 7B), they likely have low levels of nuclear back-signaling because of the lack of *Bace1*. Indeed, examination of cells expressing *Penk, Nrg1*, and *Bace1* in WT DG resulted in <10 cells

showing co-expression of all three markers, whereas most of the *Penk+Nrg1*+ GCs did not express detectable levels of *Bace1* transcripts (Fig 7C).

Thus, we conclude that the SGC-like transcriptomic state is actively repressed in the young DG by Nrg1 nuclear back-signaling and that some GCs acquire the SGC profile over adolescent development potentially because of a natural decline in Nrg1 nuclear back-signaling.

## Discussion

Efforts for cataloging brain cell types have produced hierarchical trees of hundreds of subtypes of classically recognized cell types, prompting many to ask how many bona fide cell types truly exist in the brain?; how plastic or hardwired are these cell types?; and more importantly, what bearing do they have on brain function? We suggest that a bottom-up understanding of cell-type development would aid in answering some of these questions.

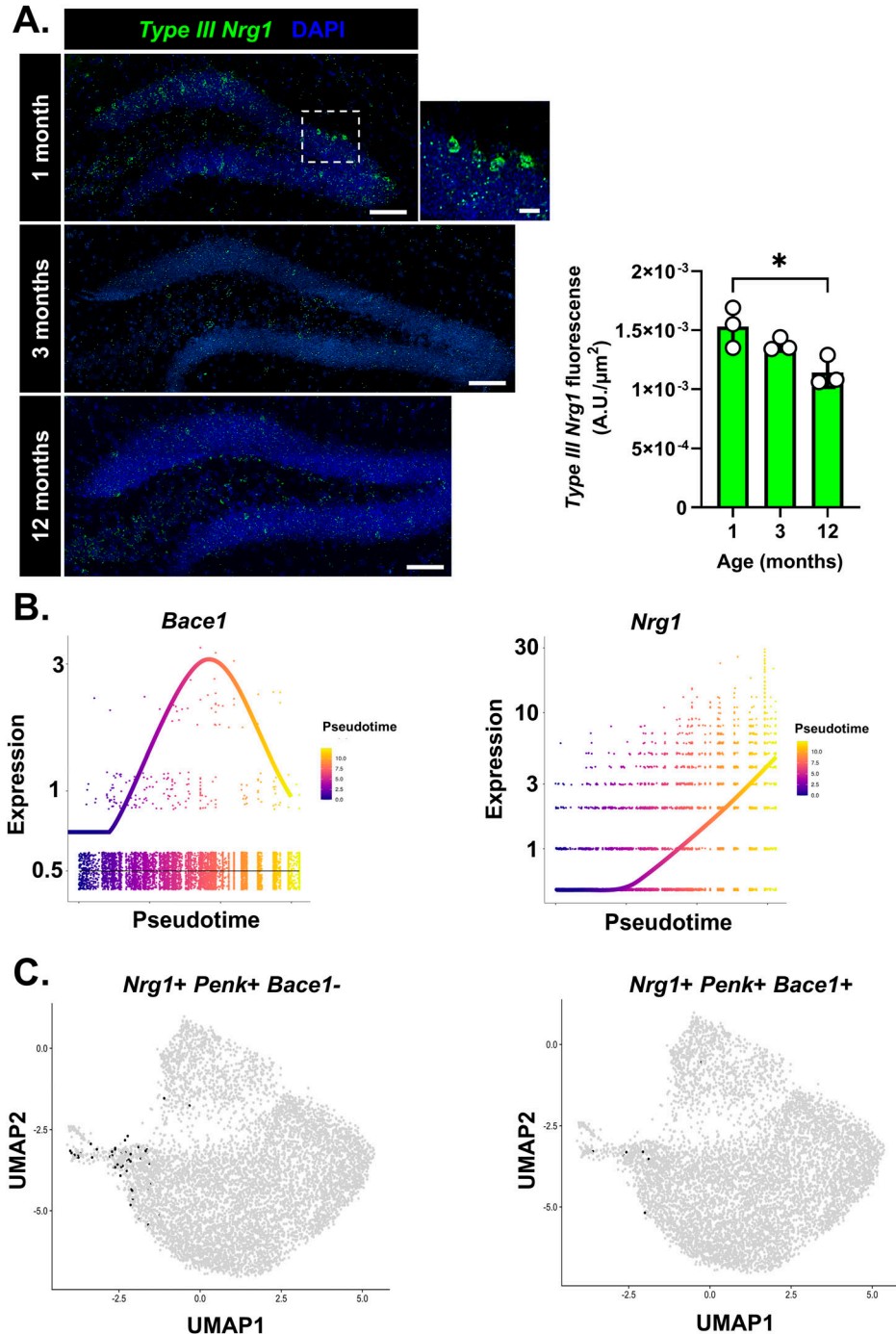

**Figure 7. Age-dependent decrease in nuclear back-signaling in the DG.**
**(A)** (Left) *Type III Nrg1* expression declines with age in the DG GCL. Representative images showing FISH for *Type III Nrg1* (green) in WT DG over age (1, 3, and 12 mo). (Right) Quantification of *Type III Nrg1* fluorescence density in the suprapyramidal blade of the DG shows a statistically significant decline in expression over age (Kruskal–Wallis test, $P$ = 0.01, KW = 6.3; Dunn's corrected multiple comparison: 1 versus 12 –mo, $P$ = 0.04). N = 3 mice/age group. **(B)** Pseudotime plots showing the expression of *Bace1* (Left) and *Nrg1* (Right) in WT GCs. *Bace1* expression peaks at intermediate pseudotime and drops at maximal pseudotime, whereas *Nrg1* expression increases along pseudotime. **(C)** UMAPs showing WT cells that express *Nrg1* and *Penk*, but not *Bace1* (Left) and cells co-expressing *Nrg1*, *Penk*, and *Bace1* (Right). Each black dot represents 1 cell expressing the designated gene(s), and gray dots represent cells not expressing the gene(s).

In this study, we investigated subtype diversity of dentate gyrus granule cells and the potential relationship of subtype composition with disease-relevant genetic variation. We found that SGCs, a morphologically distinct subtype of GCs, also showed distinct electrical properties (Fig 1). We identified unique genetic markers for SGCs and found an increase in the number of SGC marker–expressing cells with age of the animal starting between 2 and 3 mo of age (Figs 4 and 6). We assessed GC heterogeneity in a mouse model harboring a missense mutation in the *Nrg1* gene, which was previously associated with psychosis (Walss-Bass et al, 2006). There were higher numbers of morphoelectric transcriptomically defined SGCs in the mutant mouse DG even before 3 mo of age (Fig 2). An analysis of GC chromatin accessibility revealed that SGC marker genes were accessible but not expressed in WT GCs indicating a repressive mechanism preventing access to an SGC-like transcriptomic state, and potential failure of this mechanism with disease-associated genetic variation (Fig 5).

## Composition of cell-type identity: insights from the $V_{321}L$ DG

SGCs have unique morphological and intrinsic electrical properties (Fig 1). We found that all recorded neurons in the $V_{321}L$ DG, regardless of morphology, displayed SGC-like electrical properties (Fig 2I–M). We also noted that there were no major morphological distinctions between WT and $V_{321}L$ GCs or SGCs. These data indicate that the SGC fate might be driven by distinct modules of coregulated genes involved in distinct cellular functions. Trajectory analysis revealed that SGCs were among cells with the maximal pseudotime (Fig S8C and c'). Gene ontology (GO) analysis for genes whose expression was enriched in SGCs and those marking end-trajectories in the WT DG showed "regulation of RMP" as the top function represented by *Kcnj2* and *Kcnk1* (Fig S10D). A more hyperpolarized RMP was one of the SGC-specific electrical properties (Fig 1J). In $V_{321}L$ mice, we found that GCs with a single primary dendrite also showed an RMP resembling that of SGCs (Fig 2I). Upon visualizing the expression of *Kcnj2* in WT and $V_{321}L$ GC clusters, we observed that *Kcnj2* was typically (WT) expressed in cluster 18 and in cluster 1 along the pseudotime trajectory but not in clusters 0 or 5. However, in $V_{321}L$ GC clusters, *Kcnj2* expression was noted in clusters 0 and 5, which might indicate that these neurons represent the GCs with single primary dendrites and electrical properties of SGCs. A similar pattern was noted for *Kcnk1*.

Nrg1 back-signaling operates through two known mechanisms—local activation of PI3K-Akt signaling at the membrane and nuclear signaling via translocation of the ICD from the membrane to the nucleus (Bao et al, 2003; Hancock et al, 2008). The high expression of Nrg1 in SGCs with a concomitant decrease in capacity for nuclear back-signaling indicates a potential shift in the balance between local and nuclear back-signaling mechanisms (Fig 7). Enhanced Akt signaling, either via constitutive activation or loss of negative regulators like Pten and Disc1, has been shown to result in the production of GCs with multiple primary dendrites (Kim et al, 2009). Thus, Nrg1 signaling might regulate the production of the SGC phenotype through gene regulatory and local signaling mechanisms to drive global changes in electrical properties and cytoskeletal dynamics.

We noted the presence of GCs with SGC-like electrical properties (Fig 2) in the $V_{321}L$ DG. This could potentially indicate a modular switch to SGC-like electrical properties while maintaining a typical GC morphology. Faster downstroke of the action potential was found to be one of the strongest cell type–defining features of SGCs (Fig S2C). However, downstroke was one of few electrical properties which retained correlation with GC position in the GCL even after exclusion of SGCs from the analysis indicating a relationship with GC birthdate (Fig S1I). We previously noted accelerated depletion and neuronal commitment of stem cells in the adult $V_{321}L$ DG (Rajebhosale et al, 2024). It is possible that this loss of regulated neurogenesis also occurs in the late embryonic and early-postnatal stages resulting in greater production of the early-postnatal pool of GCs in $V_{321}L$ mice. Neonatally born GCs show some features of SGCs including wider splay angles as seen in the $V_{321}L$ DG (Fig 2D) (Save et al, 2019; Masachs et al, 2021). Whether these features and the additional SGC-like cells in the $V_{321}L$ DG arise from accelerated early neurogenesis and/or altered specification of postnatal and adult neurogenesis awaits further inquiry.

## The SGC: cell type or an inevitable cell state?

We found that the SGC transcriptomic fate was accessible to non-SGCs and that the numbers of cells expressing SGC markers increased with age of the animal. However, we propose that this does not indicate the presence of a single GC type representing an inevitable fate for all GCs. In the WT DG, we found that the pseudotime analysis accurately captured chronological time, ordering cells based on their birthdates (Figs 5A and B and S7C). We found only one trajectory toward the *Penk+* GCs (cluster 18), which traversed cluster 1. There was a separate trajectory ending in an intermediate pseudotime in cluster 5, with clear separation between clusters 5 and 18 (Fig 5A and B). Thus, we propose that the morphological distinction between GCs and SGCs faithfully represents two distinct types of GCs. However, there exists a third type of GC (clusters 0 and 1), which harbor the potential to acquire the SGC fate. This is apparent in the analysis of *Kcnj2* expression mentioned earlier—*Kcnj2* expression increases along pseudotime in cluster 1 but not in cluster 5 despite having identical pseudotime values. This can also be seen in the analysis of end-trajectory gene modules: clusters 1 and 18 have more common gene modules correlated with their identities, whereas cluster 5 often shows anticorrelation in those modules (Fig S10). Our analyses indicate that the access to the SGC fate might be regulated by an "epigenomic" barrier deposited by the PRC2, and erasure of this barrier and a potential concomitant activation might underlie regulation of cell-type composition in the DG (Table S4). We have previously shown that the $V_{321}L$ DG shows significantly reduced expression of *Ezh2*, which encodes the histone methyltransferase necessary for PRC2 function, and a concomitant increase in the expression of genes predicted to be PRC2 targets (Rajebhosale et al, 2024). Thus, the Nrg1 ICD might communicate developmental context such as synaptogenic interactions and/or synaptic activity to the genome through a PRC2 intermediary. Furthermore, Trajectory Geometry analyses revealed that the $V_{321}L$ GCs take a more directed trajectory potentially reflecting a loss of complexity in lineage decisions, thereby reducing the overall heterogeneity of the population. Furthermore, we also found that the open chromatin of ATAC cluster 9 cells (SGCs) was uniquely enriched for binding sites for the AP1 family of transcription factors such as Fos, Fosl, and Jun, among others (Table S6). SGCs have been shown to express high levels of Fos and as such disproportionately comprise engrams in the DG because of reliance of Fos-based labeling strategies (Erwin et al, 2020). This enrichment of AP1 binding sites might simply reflect a high level of "activity–transcription" coupling in SGCs given their higher responsivity to synaptic input. Another intriguing possibility for future investigations is that AP1 family transcription factors may play a part in mediating or maintaining the SGC fate.

Given the unique circuit properties of SGCs and their potential to release enkephalin (encoded by *Penk*), we suggest that interactions between developmentally regulated signaling and chromatin-modifying enzymes might afford the DG some flexibility to regulate cell-type composition depending on the context of the circuit, to maintain proper circuit dynamics (Larimer & Strowbridge, 2010; Gupta et al, 2020).

## Materials and Methods

### Experimental model and subject details

Male and female mice aged P24-12 mo were used. Animals were housed in a 12-h light/dark cycle environment that was both temperature- and humidity-controlled. Animals had free access to food and water. All animal care and experimental procedures were approved by the Animal Care and Use Committee of NINDS, National Institutes of Health, Bethesda MD (Animal protocol #1490).

### Genotyping

Genotypes were determined by PCR using the following primers:
Forward primer 5′-GGTGATCCCATACCCAAGACTCAG-3′.
Reverse primer 5′-CTGCACATTTATAGSGCATTTATTTTG-3′.

### Acute slice electrophysiology

Coronal brain slices containing hippocampus were prepared from P24- to P34-d-old mice. Animals were anesthetized with a mixture of ketamine and xylazine (100 mg ketamine and 6 mg xylazine/kg body weight injected ip). Then, the brains were transcardially perfused with a sucrose-based solution. After decapitation, the brain was transferred quickly into the sucrose-based solution bubbled with carbogen and maintained at ~3°C. Coronal brain slices (300 $\mu$m) were prepared using a vibratome (Leica, Inc.). Slices were equilibrated with oxygenated artificial cerebrospinal fluid (aCSF) at room temperature (24–26°C) for at least 1 h before transfer to the recording chamber.

Slices were moved to a stage of an upright, infrared differential interference contrast microscope for patch clamp recording (SliceScope, Scientifica). The recording chamber was continually perfused with aCSF to maintain a temperature of 30°C.

DG neurons were visualized with a 40 X water-immersion objective by infrared microscopy (Scientifica Pro, Scientifica). Granule cells in the middle of the GCL from the dorsal blade of the DG were recorded. In experiments focused on recording SGCs, recordings were made focused on the top cell body layer of the GCL.

Twenty features were extracted from the responses to current steps as reported in Kim et al (2024).

### Tissue processing

Animals were deeply anesthetized with isoflurane and transcardially perfused with 4% PFA in PBS. Brains were harvested and postfixed overnight at 4°C in 4% PFA in PBS. Brains were transferred to 30% sucrose in PBS, incubated at 4°C with agitation until they sank, and then embedded in optimal cutting temperature compound. After this, brains were flash-frozen and stored at –80°C. For RNAScope assays, 15-$\mu$m sections were obtained serially from Bregma –1.5 to –2.5 mm (approx). Sections were collected directly onto superfrost charged slides such that each slide contained two sections—one from a more anterior and one from a posterior hippocampal region.

### Immunohistochemistry

After acute slice recordings, 300-$\mu$m sections were fixed in 4% PFA at 4°C on a shaker overnight. The next day, sections were rinsed three times in 1xPBS and incubated in a fourth wash for 15 min at room temperature with constant agitation. Sections were then blocked with 10% normal donkey serum in 1xPBS containing 0.5% Triton X-100 at 4°C overnight on a shaker followed by incubation with fluorescently conjugated streptavidin at 4°C overnight with constant agitation. Sections were washed three times in 1xPBS containing 0.1% Triton X-100 (PBS-T) and mounted on Superfrost Plus glass slides with mounting medium containing DAPI.

### In situ hybridization

In situ hybridization was performed using the RNAScope multiplex v2 assay (ACD) or RNAScope HiPlex assay (ACD) using the manufacturer's recommended protocols available at https://acdbio.com/documents/product-documents.

Details regarding probes used can be found in the Key Resources Table (KRT; Supplemental Data 1).

### Imaging & image analysis

Imaging was done using a Nikon Ti2 spinning disk confocal microscope. All slides in an experiment were imaged using identical settings. Images were acquired by capturing the whole DG and using z-steps of 1 $\mu$m using a 40x silicon oil-immersion objective. Image analysis was performed in ImageJ (FIJI). Images were z-projected using maximum intensity projection, and regions of interest were manually outlined. Fluorescence intensity density was measured as average intensity normalized to the area of the superior blade of the DG. The cell counter plug-in was used to manually count cells expressing *Penk* and other marker genes. We counted cells expressing 5 or more small puncta of any given gene or those with large clusters of RNA molecules. For most of the probes used, the RNA expression profile was "soma filling." To analyze cell positions, we measured the distance of individual cells to the GCL-MOL boundary by measuring the length of a line drawn to be perpendicular to the GCL-MOL boundary.

Imaged neurobiotin-filled neurons were manually reconstructed and analyzed using the filaments feature in Imaris (Oxford Instruments). Splay angles were calculated in FIJI using the "Angle tool" by sequentially placing one point on the most laterally displaced dendritic termination, followed by one in the center of the soma and then on the opposite most laterally displaced dendritic termination. Cell position was recorded by finding the optical section containing a section through the center of the cell body. Vertical distance from the center of the cell body to the GCL-MOL boundary was measured, manually considering the angle of the blade of the DG.

### Single-nucleus RNA+ATAC-Seq

Dentate gyrus microdissections were performed as reported previously (Rajebhosale et al, 2024). Tissue was stored at –80°C overnight. The next day, single-nucleus suspension was prepared

from the frozen DG tissue using the Chromium Nuclei Isolation kit (10xGenomics; Cat#1000494) following the manufacturer's guidelines. Nuclei were counted and diluted appropriately to obtain a final concentration of ~1,000 nuclei/µl. ~3,000–5,000 nuclei per animal were processed for GEM generation. ATAC-Seq and RNA-Seq libraries were prepared using components of the Chromium Next GEM Single Cell Multiome ATAC + Gene Expression Reagent Bundle (10xGenomics; Cat#1000283) according to the manufacturer's guidelines. Libraries were sequenced on an Illumina NextSeq 550 (High throughput kit) with ~25,000 reads per nucleus for RNA-Seq and ~20,000 read pairs per nucleus for ATAC-Seq. The cellranger-arc-2.0.2 pipeline (10xGenomics) was then applied to the sequence output producing data files for gene expression and ATAC-Seq analysis.

### Gene expression and ATAC-Seq analysis

Sequenced libraries were analyzed in R using functions supported in the "Seurat" package unless otherwise described (Hao et al, 2024). Link to the detailed code for the analysis is provided in the supplemental methods (KRT).

#### Single-nucleus RNA-Seq analysis

The "filtered_feature_bc_matrix.h5" files produced per library were imported into R (https://cran.r-project.org/) using the "Read10X_h5" function from the "Seurat" package. These files were subsequently analyzed in R using functions supported in the "Seurat" package unless otherwise described. To convert each file into a "Seurat" object, the "CreateSeuratObject" function was used. To enumerate the percent mitochondrial gene expression per cell per object, the "PercentageFeatureSet" function was used (pattern = "^mt-"). To combine the objects into a single object, the "merge" function was used. The percent mitochondrial gene expression per cell along with the total number of detected counts and features per cell was then inspected using the "FeatureScatter" and "VlnPlot" functions. Cells were then filtered to keep only those that had (1) a minimum number of detected counts > 500, (2) a maximum number of detected counts < observed mean + 2SD = 8,916.1, (3) a minimum number of detected features > 250, (4) a maximum number of detected features < observed mean + 2SD = 3,307.526, and (5) a percent mitochondrial counts < observed mean + 1SD = 2.787,142%. The "SplitObject" function was then used to separate surviving cells per library into a list where after the "as.SingleCellExperiment" function and the "computeDoubletDensity" function from the "scDblFinder" package were applied. Cells observed to have a doublet score > respective library mean + 2SD were then filter-removed from the presplit object. The "Seurat::NormalizeData" function was then applied to this now doublet-curated object, and the "FindVariableFeatures" function (selection.method = "vst," nfeatures = 3,000) and the "ScaleData" function (vars.to.regress = "percent.mt") were sequentially applied. To determine the number of components to use in the clustering of cells, the "RunPCA" (npcs = 200) and "ElbowPlot" (ndims) functions were used and the "RunPCA" function was reapplied after using the number of components decided upon (npcs = 75). To integrate cells across libraries, the "RunHarmony" (plot_convergence = TRUE) and "RunUMAP" (reduction = "harmony," dims = 1:75) functions were used. For

clustering of cells, the "FindNeighbors" (reduction = "harmony," dims = 1:75) and "FindClusters" functions were used over a range of resolutions from 0.04 to 3.5 in steps of 0.01. Cluster results were summarized using the "clustree" package and 1.8 was selected as the optimal resolution. At this resolution, several clusters were manually collapsed, and then, the "findDoubletClusters" function from the "scDblFinder" package was used to screen for and remove doublet-enriched clusters. The "computeDoubletDensity" function was then reapplied but this time without splitting the cells by library. Cells observed to have a doublet score > cross-cell mean + 2SD were then filter-removed, and the "FindNeighbors" and "FindClusters" functions were reapplied under the same conditions described. Cluster results were again summarized using the "clustree" package, and 2.1 was selected as the optimal resolution. At this resolution, several clusters were again manually collapsed, and then, cells for each cluster were z-scored using UMAP dimension 1 values and separately using UMAP dimension 2 values. Cells in z-score space were then plot-inspected by cluster using the "DimPlot" function and thresholds defined and applied to filter-remove cells having a z-score greater than the defined thresholds. Annotation of the resulting z-score curated clusters was accomplished using the "RunAzimuth" function from the "Azimuth" package in conjunction with the "mousecortexref.SeuratData" reference dataset. To identify conserved markers across library class (WT and V321L), the "FindConservedMarkers" function was used. To identify nonconserved markers per cluster, the "Find-Markers" function was used. To identify differential features between library class within cluster, the "FindMarkers" function was used setting the "ident" parameters to "WT" and "V321L," respectively. Clusters annotated as "Dente gyrus" were next subset and trajectory-fit using the "Monocle3" workflow. Root selection was accomplished by inspecting the expression for "Camk4" and "Ntng1" using the "plot_cells" function. The "graph_test" function (neighbor_graph = "principal_graph") was used to identify features that explain pseudotime (q_value < 0.05). To identify depleted versus enriched modules of features that explain pseudotime, the "find_gene_-modules" function was used.

#### Single-nucleus ATAC-Seq analysis

The "atac_peaks.bed," "barcode_metrics.csv," and "atac_fragments.tsv.gz" files produced for the "LL" and "VV" dentate gyrus samples were imported into R (https://cran.r-project.org/) using the "read.table" function. Functions available in the "Signac" and "Seurat" packages were interchangeably used for analysis. To create a working fragment object per sample, the "CreateFragmentObject" function was used. Cells with an "atac_fragments" value <= 500 were discarded with surviving cells used as input into the "FeatureMatrix," "CreateChromatinAssay," and "CreateSeuratObject" functions, producing one "Seurat" object per sample. For each of these objects, genomic annotations for "EnsDb.Mmusculus.v79" were added and quality metrics generated. To add the annotations, the "GetGRangesFromEnsDb" and "Annotation" functions were used. To generate the quality metrics, the "FractionCountsInRegion," "NucleosomeSignal," and "TSSEnrichment" functions were used. Metrics produced from these functions, along with those for "peak_region_fragments," "nCount," "pct_reads_in_peaks," and "TSS_fragments," were used to filter-remove cells from each object via the "subset" function.

Specifically, cells that did not have a value within 2SD of the observed mean for each metric were removed. Post-filtering, all "Seurat" objects were combined into one using the "merge" function and a unified set of peaks generated using the "reduce" function. This unified peak set was then filtered to keep only those with a "peakwidths" value < 10,000 and > 20. The filtered peak set was then used to regenerate the individual "Seurat" objects as described, and those objects were combined into one object. The "RunTFIDF," "FindTopFeatures," "RunSVD," "RunUMAP," and "Find-Neighbor" functions were then applied to the combined object (dims = 2:20, reduction = "lsi") in preparation for clustering. For clustering, the "FindClusters" function was used over a range of resolutions from 0.04 to 3.5 in steps of 0.01. Cluster results were summarized using the "clustree" package, and 1.5 was selected as the optimal resolution. At this resolution, several clusters were manually collapsed, and then, cells for each cluster were z-scored using UMAP dimension 1 values and separately using UMAP dimension 2 values. Cells in z-score space were then plot-inspected by cluster using the "DimPlot" function and filter thresholds defined. Cells having a z-score greater than the thresholds were removed. For surviving cells, annotations were transferred from the single-nucleus RNA-Seq analysis and log-normalized activities for each known gene calculated. For annotation transfer, the "FindVariableFeatures" function (nfeatures = 5,000) was used in conjunction with the "TransferData" function (reduction = "cca," weight.reduction = "lsi," dims = 2:20). To calculate the log-normalized gene activities, and to add them to the combined "Seurat" object, the "GeneActivity," "CreateAssayObject," and "NormalizeData" functions were used. Specific clusters representing <HERE> were then subset and the functions used prior, per preparation for clustering, clustering, and post-clustering, used again. For this subset, 0.5 was selected as the optimal resolution. Differential peaks between clusters produced at this resolution were tested for using the "FindMarkers" function (test.use = "LR"). Overrepresented motifs available in the "JASPAR2020" database (collection = "CORE," tax_group = "vertebrates") were also tested for using the "FindMotifs" function. Post-testing, results were annotated with the closest occurring known gene using the "ClosestFeature" function. In addition to this testing, trajectory fitting was performed using the "Monocle3" workflow. This workflow was applied in three separate tacts: (1) using all cells representing "VV" and "LL" samples combined, (2) using cells for "VV" samples only, and (3) using cells for "LL" samples only. Root selection for each of these tacts was accomplished by inspecting the gene activities for "Camk4," "Igfbpl1," "Fxyd7," "Dcx," "Calb2," "Calb1," "Ntng1," "Penk," and "Sorcs2" using the "plot_cells" function. Post-fitting, the "cor.test" function was used in conjunction with the "P.adjust" function (method = "BH") to test for and identify genes having pseudotime correlated with gene activity (q_value < 0.05). Enriched terms for these genes were subsequently identified using the "enrichR" package (dbs = "GO_Molecular_Function_2023," "GO_Cellular_Component_2023," "GO_Biological_Process_2023").

### Trajectory Geometry analysis

Trajectory directionality was assessed using the TrajectoryGeometry R package. For each condition (WT downsampled, WT original, and $V_{321}L$), PCA embeddings (top 50 principal components) and UMAP-derived pseudotime values were extracted from Monocle3 trajectory-fitted single-cell objects. PCA coordinates were rescaled to a 0–100 range to ensure comparability across datasets. The analyseSingleCellTrajectory() function was run with 1,000 sampling windows (nSamples = 1,000) and 1,000 randomized path permutations (N = 1,000) using the randomizationParams = c("byPermutation," "permuteWithinColumns") setting to assess statistical significance of trajectory directionality based on spherical distances. The test statistic was set to "mean" to capture average directional consistency.

## Statistical analysis

Statistical analysis was performed using Prism (GraphPad). Normality was assessed using the Shapiro–Wilk and the Kolmogorov–Smirnov tests. If data failed normality test, nonparametric statistics were used. P-values were corrected for multiple comparisons as necessary using Bonferroni's (parametric) and Dunn's (nonparametric) post hoc tests. Multiple linear regression was performed in R using the *lm* base R function to create the following linear model: $\beta 0 + \beta 1 \cdot \text{Cell Type} + \beta 2 \cdot \text{Soma position} + \beta 3 \cdot (\text{Cell Type} \times \text{Soma position}) + \epsilon$.

### *LDA*

Before analysis, all numerical features were standardized using z-score normalization, where each feature was transformed to have a mean of 0 and a SD of 1. Standardization was performed using the *scale()* function from base R. To reduce dimensionality and mitigate potential multicollinearity among features, PCA was performed using the prcomp() function. The number of principal components (PCs) used was dynamically adjusted to the minimum number available across both WT and $V_{321}L$ datasets, ensuring consistent dimensionality. For the WT dataset, the top 9 PCs were retained and used to train an LDA model using the train() function from the caret package with fivefold cross-validation (trainControl [method = "cv," number = 5]), and the lda method from the MASS package. The trained PCA and LDA models were then applied to the $V_{321}L$ dataset. The same PCA transformation (from WT) was applied to the scaled V321L data using predict(), and the top 9 PCs were extracted. Predicted class labels were generated using the trained LDA model (predict() from MASS). Model performance on the $V_{321}L$ dataset was assessed using a confusion matrix (confusionMatrix() from caret), comparing predicted versus true labels. The confusion matrix was visualized using ggplot2, with cell frequency values represented by both color intensity and overlaid counts.

# Data Availability

The data generated in this publication have been deposited in NCBI's Gene Expression Omnibus (Edgar et al, 2002) and are accessible through GEO Series accession numbers provided in the KRT. Code for data analysis can be accessed on GitHub using the links provided in the KRT.

# Supplementary Information

# Acknowledgements

This work was supported by Intramural Research Program of NINDS to DA Talmage (1ZIANS009424) and LW Role (1ZIANS009416). We would like to thank Dr. Abdel Elkahloun and Bayu Sisay (NHGRI Sequencing Core Facility) for providing NGS consultation and services. In addition, we would like to thank Taylor Muir for mouse colony management and Li Bai, Jessie Wallace, Shaina Sindhu, and Becca Alvarez for technical assistance with project management.

## Author Contributions

P Rajebhosale: conceptualization, data curation, software, formal analysis, investigation, visualization, methodology, and writing—original draft, review, and editing.
L Jiang: investigation.
HJ Ressa: investigation.
KR Johnson: data curation, software, formal analysis, visualization, and methodology.
NS Desai: software and formal analysis.
A Jone: investigation.
LW Role: conceptualization, supervision, funding acquisition, project administration, and writing—review and editing.
DA Talmage: conceptualization, resources, supervision, funding acquisition, project administration, and writing—review and editing.

## Conflict of Interest Statement

The authors declare that they have no conflict of interest.

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
