## [Reviewer comments · Life Science Alliance]

Life Science Alliance

Diversification of Dentate Gyrus Granule Cell Subtypes is Regulated by Nrg1 Nuclear Back Signaling

Prithviraj Rajebhosale, Li Jiang, Haylee J. Ressa, Kory R. Johnson, Niraj S. Desai, Alice Jone, Lorna W. Role, and David A. Talmage

DOI: <https://doi.org/10.26508/lsa.202403169>

Corresponding author(s): David Talmage, National Institute of Neurological Disorders and Stroke

Review Timeline:

Submission Date:	2024-12-10
Editorial Decision:	2025-03-07
Revision Received:	2025-03-26
Editorial Decision:	2025-04-11
Revision Received:	2025-04-12
Accepted:	2025-04-16

Scientific Editor: Tim Fessenden

Transaction Report:

March 7, 2025

Re: Life Science Alliance manuscript #LSA-2024-03169-T

David A Talmage
National Institute of Neurological Disorders & Stroke

Dear Dr. Talmage,

Thank you for submitting your manuscript entitled "Diversification of Dentate Gyrus Granule Cell Subtypes is Regulated by Neuregulin1 Nuclear Back Signaling." to Life Science Alliance. The manuscript was assessed by expert reviewers, whose comments are appended to this letter. We invite you to submit a revised manuscript addressing the Reviewer comments.

Thank you for this interesting contribution to Life Science Alliance. We are looking forward to receiving your revised manuscript.

Sincerely,

B. MANUSCRIPT ORGANIZATION AND FORMATTING:

Reviewer #1 (Comments to the Authors (Required)):

In the study "Diversification of Dentate Gyrus Granule Cell Subtypes is Regulated by Neuregulin1 Nuclear Back Signaling," Rajebhosale and colleagues focus on two types of GCs (typical and semilunar), providing a detailed characterization of these cell types, and how their abundance, electrophysiology, and spatial location change with a disease-associated genetic mutation (V321L). They then provide a plausible mechanism for their findings and show that similar changes occur in normal aging. This study advances previous work by showing that GCs can directly transition into SGCs and that this transition may be suppressed by nuclear back propagation, which has previously been associated with adult neurogenesis. Furthermore, the multimodal characterization of these cell subtypes with morphology, electrophysiology, transcriptomics, epigenomics, and spatial profiling in both healthy and disease mice is extensive.

This is a well-written paper that clearly describes the problem being addressed in the context of what is currently known about GC development, and then presents a series of experiments and associated analysis to compellingly address the problem. There are a few minor issues laid out below, but the claims are well-supported overall, even for a non-expert.

Minor comments:

1. Please clarify why this specific mutation was chosen for this paper. Is it based on previous work, because a mutant mouse is available, and/or for other reasons?
2. Define all acronyms at first usage (e.g., NSC and SGZ are not defined).
3. Quantify this statement in the discussion of figure 2: "Additionally, GCs from V321L mice showed wider splay angles than WT GCs", as this seems to be a key point that comes up again in the Discussion. Figure 2D currently shows the difference between GCs and SGCs, but not between wild type and mutant mice.
4. In Figure 2G it looks like the SGCs in the wild type mice are constrained but the mutant mice represent a larger and somewhat distinct space in LDA space (e.g., they are below the wild type SGCs overall). If this is not statistically significant, please add a statement to that effect.
5. Provide better explanation about why Suz12 regulation was assessed. Based on Table S3, it appears to be the most significant of the four terms shown, but what method was used, and where do these terms come from? Are there other terms that were tested but failed to reach significance, or was this a supervised test only looking for terms associated with Nrg1 and nuclear back propagation?
6. Having huge odds ratios based on expression of only two genes, especially genes that are related (e.g., Figure S8D) is not convincing by itself, although alignment of these genes with actual RMP effects from electrophysiology mitigates this concern a bit. Do other genes in the regulation of the RMP pathways also show similar trends, even if not reaching the level of significance?
7. There are a few typos in the paper to address (e.g., Figure 8E/F  Figure S8E/F)
8. Why only follow up with Nrg1 in Figure 7? Is it from previous studies? Is it because Nrg1 is related to nuclear back-signaling? Would mutation in any other marker genes from Figure 4 show the same results?
9. The code is hard to follow, as presented in a few very long files, and if possible, it would be useful to split this out. That said, inclusion of the code on GitHub for reproducibility is much appreciated, and the existing code annotations make it usable in its current format if needed.

Reviewer #2 (Comments to the Authors (Required)):

I have reviewed the manuscript by Rajebhosale et al., in which the authors investigate the characteristics of a morphological subtype of granule cells in the dentate gyrus of the hippocampus, providing further evidence for their distinct molecular and electrophysiological features. The study is well-reasoned, the research questions are relevant, and the structure is sound and well-organized. However, several methodological aspects require further attention to strengthen the robustness of some conclusions and improve the reproducibility of certain analyses.

Major comments:

1. The study initially focuses on characterizing the differences between semilunar granule cells (SGCs) and granule cells (GCs). While SGCs are primarily located in superficial layers, suggesting an earlier developmental origin, some electrophysiological properties distinguishing SGCs from GCs also covary with layer depth among GCs, indicating a potential dependence on GC age. A more detailed statistical analysis is recommended, incorporating a full model that assesses the contribution of the SGC/GC condition to each electrophysiological feature while controlling for soma position. The same model could then be applied to the V321L mouse.

2. A key question in this study is whether SGCs represent a distinct cell type within the dentate gyrus or merely a specific maturation state of GCs that can be promoted by enhanced differentiation (e.g., via NRG1 mutations). The authors appropriately address this question using single-cell analysis. One cluster appears enriched in SGCs, as indicated by the expression of *Penk*, a gene previously associated with GCs that are morphologically and developmentally similar to SGCs. This cluster also exhibits selective expression of *Sorcs3*, which may serve as a putative SGC marker. Given that *Penk* and *Sorcs3* colocalization increases in the V321L mouse model, where SGC numbers are also elevated throughout the DG, this result is expected. However, quantifying the proportion of cells expressing *Penk* only, *Sorcs3* only, or both would help clarify the specificity of *Sorcs3* as an SGC marker.

3. The single-cell analysis is critical for defining differentiation trajectories. However, the methods section lacks details on data analysis, integration methods, and parameter settings. A comprehensive description of these aspects should be included.

4. The authors argue that the existence of distinct differentiation trajectories leading to terminal GC types supports the idea that SGCs represent a unique cell type rather than a terminal state that any GC can achieve. However, the cell clusters traversed in WT and V321L mice are not balanced across donors. A downsampling strategy could help ensure a more accurate comparison of lineage reconstructions.

5. The proposed involvement of PRC2 in repressing the SGC phenotype is not well substantiated. More details on the ChEA/ENCODE analysis or alternative approaches should be provided. For instance, a permutation-based approach could assess the likelihood of PRC2 targets being enriched in random sets of GC-expressed genes of similar size (e.g., 60 genes).

6. Investigating recently born SGCs to identify signatures unrelated to cell maturation is a valuable approach. The authors describe *Penk* expression and other SGC-enriched genes, but it would be informative to examine the expression of *Sorcs3* and *Nptx2* in these early SGCs as well.

Minor comments:

In Figure 2A, some boxes of the boxplot are colored black and medians cannot be distinguished.

Re: #LSA-2024-03169-T

"Diversification of Dentate Gyrus Granule Cell Subtypes is Regulated by Neuregulin1 Nuclear Back Signaling."

We are pleased that the editors and reviewers found our manuscript of interest and appreciate their thoughtful recommendations. In response, we have conducted additional analyses and revised the text and figures accordingly. Below, we provide our point-by-point responses to the reviewers' comments, with our responses highlighted in **blue**. References to figures in the revised manuscript are indicated as "**New Figure**".

Response to Reviewer #1

We thank the reviewer for their kind words and helpful suggestions. In response we have made the following changes to address their concerns.

Minor comments:

1. Please clarify why this specific mutation was chosen for this paper. Is it based on previous work, because a mutant mouse is available, and/or for other reasons?

The choice of studying the Nrg1 V₃₂₁L mutation was motivated by three main reasons:

1. In our characterization of SGCs in WT mice, we found that expression of Nrg1 was enriched in SGCs relative to typical GCs indicating a potential relevance of Nrg1 signaling to SGC biology (**Figure 4B, b'**).
2. In our previous study we found that the V₃₂₁L mutation in Nrg1 altered the differentiation of neural stem cells and altered dendritic complexity of granule cells (GCs) in the DG (Rajebhosale, Jone et al. 2024). We performed Golgi staining to assess dendritic complexity of GCs in WT and V₃₂₁L mutant mice, and in a follow-up, we noted the overabundance of SGC-like cells in the mutant mice (shown in **Figure 2A**).
3. The Nrg1 V₃₂₁L mutation was initially identified in an isolated Costa Rican population as being associated with psychosis and schizophrenia. Several other animal models and patient postmortem samples from patients with neuropsychiatric disorders (especially schizophrenia) have shown the overabundance of cells with a similar morphology (Lauer, Beckmann et al. 2003, Kim, Duan et al. 2009, Fitzsimons, van Hooijdonk et al. 2013, Llorens-Martin, Jurado-Arjona et al. 2014, Terreros-Roncal, Flor-Garcia et al. 2019, Marquez-Valadez, Rabano et al. 2022); as a result we found it pertinent to test the hypothesis that these might represent SGCs in the V₃₂₁L mutant mouse.

We attempted to incorporate these points as a primary finding in Figure 4 (Point 1), and in the second half of the introduction (points 2 and 3).

Define all acronyms at first usage (e.g., NSC and SGZ are not defined).

We thank the reviewer for their attention to this. **Line 130** now defines NSC. SGZ is defined in **Line 68** of the introduction.

2. Quantify this statement in the discussion of figure 2: "Additionally, GCs from V321L mice showed wider splay angles than WT GCs", as this seems to be a key point that comes up again in the Discussion. Figure 2D currently shows the difference between GCs and SGCs, but not between wild type and mutant mice.

Unlike the statistical comparison between WT GCs and SGCs, the comparison of splay angles of V₃₂₁L GCs and SGCs did not achieve the p-value criterion for significance (Figure 2D). However, there were no statistically significant differences between WT and V₃₂₁L GCs (**p=0.92**) either.

We have now corrected the text to omit the direct comparison of WT GCs and V₃₂₁L GCs in the results and discussion sections.

Lines 199-200: "*There were no significant differences in the splay angle between GCs and SGCs (Figure 2D, $p=0.08$).*"

Lines 562-563: "*We also noted that there were no major morphological distinctions between WT and V₃₂₁L GCs or SGCs.*"

3. In Figure 2G it looks like the SGCs in the wild type mice are constrained but the mutant mice represent a larger and somewhat distinct space in LDA space (e.g., they are below the wild type SGCs overall). If this is not statistically significant, please add a statement to that effect.

In a linear discriminant analysis (LDA), LD1 acts as the primary classifier for the two groups. LD2 is not a good projection to assess separation between groups. The different spread of datapoints along LD2 might indicate some change in variance within each dataset however, it is not predictive of cell type.

In accordance with the reviewer's suggestion, we have added this clarification in the results section.

Lines 205-207: "*LDA resulted in successful capture of cell type differences between GCs and SGCs (along the primary classifier- LD1) but not between genotypes (Figure 2G)*"

Lines 212-216: "*We did note a differential spread of WT and V₃₂₁L SGCs along LD2 which might indicate a potential change in variance within each dataset however, given the poor functionality of LD2 projection as a classifier, these data indicate that although the SGC-like cells were over-represented and mis-localized in the V₃₂₁L, morphological distinctions between GCs and SGCs were preserved (Figure 2H).*"

4. Provide better explanation about why Suz12 regulation was assessed. Based on Table S3, it appears to be the most significant of the four terms shown, but what method was used, and where do these terms come from? Are there other terms that were tested but failed to reach significance, or was this a supervised test only looking for terms associated with Nrg1 and nuclear back propagation?

Indeed, other terms were tested but failed to reach significance. The method used to calculate p-values for these transcriptional regulators was the Fisher's exact test and the p-values were adjusted using the Benjamini-Hochberg method to correct for multiple testing. This method was used to query the ENCODE & CHEA datasets using Enrichr which has a library of 104 terms (transcriptional

regulators) covering over 15,000 genes in the genome. We set a threshold of adjusted $p < 0.05$ to filter out transcriptional regulators that were not significantly enriched leaving us with the 4 remaining candidates. Of these candidates Suz12 had the smallest adjusted p-value and highest numbers of genes that were predicted to be regulated compared to the other three terms. Furthermore, in a previous study, we found that core components of the polycomb repressive complex 2 (PRC2) were downregulated in the $V_{321}L$ DG (Rajebhosale, Jone et al. 2024).

5. Having huge odds ratios based on expression of only two genes, especially genes that are related (e.g., Figure S8D) is not convincing by itself, although alignment of these genes with actual RMP effects from electrophysiology mitigates this concern a bit. Do other genes in the regulation of the RMP pathways also show similar trends, even if not reaching the level of significance?

We acknowledge the reviewer's concern regarding the high odds ratio resulting from the enrichment of only two genes from the Regulation of Resting Membrane Potential (RMP) GO term. This GO term consists of 17 genes, including 8 from the *Kcnk* family and 2 from the *Kcnj* family of potassium channels. Our tested gene set comprised 60 genes, obtained as the intersection of two sets: genes enriched in SGCs and genes correlating with pseudotime trajectories leading to cluster 18 (SGC-enriched cluster). Given that only 17 genes populate the RMP GO term, the inclusion of 2 of these genes in a 60-gene set resulted in a high odds ratio.

In addition to this statistical enrichment, experimental evidence supports our findings. Previous studies have demonstrated that augmenting *Kcnj2* function in neurons is sufficient to alter the resting membrane potential by a magnitude comparable to the difference observed between GCs and SGCs ($\sim 10\text{mV}$) (Auffenberg, Jurik et al. 2016). This alignment between our enrichment analysis, electrophysiological data, and trajectory analysis strengthens confidence in this result.

Regarding the reviewer's question, we examined whether additional genes from the RMP GO term were represented in either of the two sets that formed our 60-gene intersection. While we found no other members present, we believe that the integration of multiple lines of evidence provides a compelling case for the observed findings.

6. There are a few typos in the paper to address (e.g., Figure 8E/F  Figure S8E/F)

We apologize for this oversight and appreciate the reviewer's attention to this. It has now been corrected as **New Figure S9E/F** as we have added a supplemental figure during revision of this manuscript.

7. Why only follow up with *Nrg1* in Figure 7? Is it from previous studies? Is it because *Nrg1* is related to nuclear back-signaling? Would mutation in any other marker genes from Figure 4 show the same results?

The decision to focus on *Nrg1* in Figure 7 was driven by the need to resolve an apparent contradiction in our findings: (1) high *Nrg1* expression in SGCs and (2) an increase in SGCs resulting from a *Nrg1* loss-of-function mutation. With this analysis, we aimed to clarify this paradox.

We found that the *Nrg1* $V_{321}L$ mutation, which disrupts nuclear back-signaling, led to an overabundance of SGCs, suggesting that this signaling pathway may act to suppress SGC fate.

Consistent with this, the increase in SGC marker+ cells in the DG of V₃₂₁L mice was observed specifically in *Nrg1*-expressing cells (**New Figure S6**). Given that *Nrg1* is expressed at relatively higher levels in SGCs than in typical GCs, we hypothesized that in WT mice, despite high *Nrg1* expression, SGCs may have lower levels of *Bace1*, the enzyme required to generate nuclear back-signaling-competent *Nrg1*. Indeed, we found that almost no SGCs co-expressed *Nrg1* and *Bace1*, suggesting that *Nrg1* in WT SGCs is likely not capable of nuclear back-signaling.

Further supporting this hypothesis, we observed that the Type III isoform of *Nrg1*, which is nuclear back-signaling competent, showed a gradual decline in expression levels during postnatal development, aligning with the timeframe in which SGC numbers progressively increased (Figure 7A).

Regarding the reviewer's question about whether mutations in other SGC marker genes from Figure 4 would produce similar results, this is indeed an intriguing question for future studies. While we did not test this directly, our findings with *Nrg1* provide a mechanistic framework that could be extended to explore the roles of other genes in regulating SGC fate.

8. The code is hard to follow, as presented in a few very long files, and if possible, it would be useful to split this out. That said, inclusion of the code on GitHub for reproducibility is much appreciated, and the existing code annotations make it usable in its current format if needed.

We appreciate the reviewer's feedback and the acknowledgment of our efforts to ensure reproducibility. To improve accessibility and usability, we have updated the README file to include a structured guide outlining the key sections of the code:

README Lines 1–1960: snRNA-Seq preprocessing and clustering

README Lines 1962–2946: Analysis of gene expression in WT and V321L mice

README Lines 2948–3228: Trajectory analysis

README Lines 3278–4775: snATAC-Seq preprocessing and clustering

README Lines 4777–6667: Analysis of differential accessibility in WT and V321L mice

We hope that this guide will allow users to more easily locate the relevant code components. We recognize the benefit of further modularizing the code and will consider restructuring it into smaller files in future updates.

(Response to Reviewer 2 continued on next page)

Response to Reviewer #2

We thank the reviewer for their positive evaluation of our work and for providing constructive feedback to further enhance the quality of our findings. In response we have made the following changes to address the concerns listed by them.

Major comments:

1. The study initially focuses on characterizing the differences between semilunar granule cells (SGCs) and granule cells (GCs). While SGCs are primarily located in superficial layers, suggesting an earlier developmental origin, some electrophysiological properties distinguishing SGCs from GCs also covary with layer depth among GCs, indicating a potential dependence on GC age. A more detailed statistical analysis is recommended, incorporating a full model that assesses the contribution of the SGC/GC condition to each electrophysiological feature while controlling for soma position. The same model could then be applied to the V321L mouse.

The reviewer's point is well-taken that excluding SGCs from linear regression when analyzing individual properties (**Figure S1**) may not be the most robust approach to delineate cell type effects. We implemented a Multiple Linear Regression (MLR) model that includes both cell type and soma position as factors and applied this model to the V₃₂₁L dataset to assess whether GC-SGC distinctions were preserved.

In WT mice, spike amplitude and upstroke significantly differentiated GC and SGC, even after controlling for soma position. However, in V₃₂₁L mice, none of the features that explained cell type differences in WT remained significant, supporting our conclusion that the V₃₂₁L mutation disrupts electrophysiological distinctions between GC and SGC.

To further explore this, we conducted multivariate analyses. We trained a Linear Discriminant Analysis (LDA) classifier using WT data and applied it to classify V₃₂₁L cells based on electrophysiological properties. The classifier achieved 52% accuracy, indicating a marked reduction in resolution between GC and SGC in V₃₂₁L mice. Additionally, centroid analysis along the first linear discriminant axis (LD1) demonstrated reduced separation of GC and SGC centroids in the V₃₂₁L dataset, further confirming the blurring of cell-type distinctions.

Together, these results support the conclusion that the V₃₂₁L mutation disrupts GC-SGC distinctions, not only at the level of individual properties but also in their multivariate electrophysiological profile.

We have edited the text in the results and methods sections and added plots to indicate these additional analyses.

Lines 230-252: “V321L mice exhibited mis-localized SGCs (Figure 2H). In WT mice, several electrophysiological properties co-varied with soma position (Figure S1). To account for both soma position and cell type, we applied a multiple linear regression model to WT mice, then extended this model to V321L mice (see methods for details).

In WT mice, we found that two electrophysiological properties, spike amplitude ($p = 0.042$, $\beta_1 = -46.2$) and upstroke ($p = 0.038$, $\beta_1 = -117.88$), significantly covaried with cell type when controlling for soma position. However, this model did not detect significant cell type effects for any electrophysiological properties in V321L mice.

Given the large number of electrophysiological parameters measured (20) and their potential correlations, we further used a multivariate approach to distinguish cell types based on electrical properties. Principal component analysis (PCA) revealed a clear separation between GCs and SGCs in WT mice along the first two principal components, whereas cells in V321L mice showed substantial overlap (Figure S2E).

To further assess classification accuracy, we trained an LDA classifier with 5-fold cross validation on WT electrophysiological data and applied it to V321L mice. Centroid analysis demonstrated a striking reduction in the separation between GC and SGC centroids in V321L mice along the first linear discriminant (LD1) axis (Figure S2F). The LDA classifier, trained on 80% of the WT data, achieved 78% accuracy in cell type classification of the remainder 20% WT cells (Cohen's $\kappa=0.57$) (Figure S2G Top). Using this classifier on the recordings obtained from V321L mice resulted in 52% classification accuracy (Cohen's $\kappa=0.08$) (Figure S2G Bottom). Thus, these data further support the finding of a loss of electrophysiological distinction between GCs and SGCs in the V321L mutant DG.”

Also see New Figure S2 E-G.

Lines 857-876: “Multiple linear regression was performed in R using the *lm* base R function to create the following linear model: $\beta_0 + \beta_1 \cdot \text{Cell Type} + \beta_2 \cdot \text{Soma position} + \beta_3 \cdot (\text{Cell Type} \times \text{Soma position}) + \epsilon$. Linear Discriminant Analysis: Prior to analysis, all numerical features were standardized using z-score normalization, where each feature was transformed to have a mean of 0 and a standard deviation of 1. Standardization was performed using the *scale()* function from base R. To reduce dimensionality and mitigate potential multicollinearity among features, Principal Component Analysis (PCA) was performed using the *prcomp()* function. The number of principal components (PCs) used was dynamically adjusted to the minimum number available across both WT and V₃₂₁L datasets, ensuring consistent dimensionality. For the WT dataset, the top 9 PCs were retained and used to train a Linear Discriminant Analysis (LDA) model using the *train()* function from the *caret* package with 5-fold cross-validation (*trainControl*(method = "cv", number = 5)), and the *lda* method from the *MASS* package. The trained PCA and LDA models were then applied to the V₃₂₁L dataset. The same PCA transformation (from WT) was applied to the scaled V321L data using *predict()*, and the top 9 PCs were extracted. Predicted class labels were generated using the trained LDA model (*predict()* from *MASS*). Model performance on the V₃₂₁L dataset was assessed using a confusion matrix (*confusionMatrix()* from *caret*), comparing predicted versus true labels. The confusion matrix was visualized using *ggplot2*, with cell frequency values represented both by color intensity and overlaid counts.”

2. A key question in this study is whether SGCs represent a distinct cell type within the dentate gyrus or merely a specific maturation state of GCs that can be promoted by enhanced differentiation (e.g., via NRG1 mutations). The authors appropriately address this question using single-cell analysis. One cluster appears enriched in SGCs, as indicated by the expression of *Penk*, a gene previously associated with GCs that are morphologically and developmentally similar to SGCs. This cluster also exhibits selective expression of *Sorcs3*, which may serve as a putative SGC marker. Given that *Penk* and *Sorcs3* colocalization

increases in the V321L mouse model, where SGC numbers are also elevated throughout the DG, this result is expected. However, quantifying the proportion of cells expressing *Penk* only, *Sorcs3* only, or both would help clarify the specificity of *Sorcs3* as an SGC marker.

We appreciate the reviewer's insight regarding the specificity of *Sorcs3* as an SGC marker. To address this, we quantified the proportion of *Sorcs3*-expressing but *Penk*-negative GCs in both WT and V₃₂₁L mutant mice. Our analysis revealed no significant difference in the percentage of these cells between the two genotypes, suggesting that the mutation does not broadly affect *Sorcs3* expression outside of *Penk*+ populations.

Additionally, we examined the proportion of *Penk*-expressing but *Sorcs3*-negative GCs. While the overall percentage of *Penk*+ cells (*Sorcs3*+ or -) remained stable in WT mice, V₃₂₁L mutants exhibited an increase in both *Penk*+*Sorcs3*+ and *Penk*+*Sorcs3*- populations.

Importantly, the fact that the proportion of *Penk*-*Sorcs3*+ cells remained unchanged supports the specificity of *Sorcs3* as an SGC marker, as its expression is not broadly upregulated across all GCs. However, the expansion of *Penk*+*Sorcs3*- cells raises interesting questions regarding potential heterogeneity within the SGC subtype. Future studies are needed to determine whether *Penk*+*Sorcs3*- and *Penk*+*Sorcs3*+ neurons represent distinct functional populations or exist along a dynamic continuum.

In support of this distinction, we found that while *Penk*+ cells were present in Cluster 18 (fewer cells but a higher percentage of the cluster) and Cluster 1 (more cells but a lower percentage of the cluster), *Penk*+*Sorcs3*- cells were excluded from Cluster 18 in both WT and V₃₂₁L mice. This suggests that *Penk*+*Sorcs3*- cells may define a specific subset of SGCs with distinct clustering properties.

We have added text and a figure (**New Figure S4**) to further explore this point in the manuscript.

Lines 283-296: *"To further assess the specificity of Sorcs3 as an SGC marker in WT and V₃₂₁L mice we quantified the proportions of (Penk+Sorcs3+), (Penk+Sorcs3-), and (Penk-Sorcs3+) GCs in both WT and V₃₂₁L mutant mice. We found that in line with the increase in proportion of Penk+ cells (Figure 3D), V₃₂₁L mice also showed an increase in the percentage of Penk+Sorcs3+ GCs (Figure S4A; p=0.009). We found that there were no significant differences in the proportion of Penk-Sorcs3+ cells between WT and V₃₂₁L mice (Figure S4A; p=0.29) indicating that the mutation does not affect Sorcs3 expression outside the Penk+ population of GCs indicating its retained specificity in the mutant mice. However, upon examining the proportions of Penk+Sorcs3- GCs, we found that this population of GCs was also expanded in the V₃₂₁L mutant mice (Figure S4A; p=0.0002) indicating the potential presence of further heterogeneity within the Penk+ (SGC) population. Upon examination of clustering of these subpopulations, we found that Penk+Sorcs3- cells were excluded from cluster 18 in both WT and V₃₂₁L mice (Figure S4B&C)."*

3. The single-cell analysis is critical for defining differentiation trajectories. However, the methods section lacks details on data analysis, integration methods, and parameter settings. A comprehensive description of these aspects should be included.

We provide the requested details in the methods section. In addition, we also provide a readme file with the code which provides line number annotations for the code such that subsections pertaining specific analyses can be more readily accessed.

Lines 734-850: “Single-nucleus RNA-Seq Analysis: The "filtered_feature_bc_matrix.h5" files produced per library were imported into R (<https://cran.r-project.org/>) using the "Read10X_h5" function from the "Seurat" package. These files were subsequently analyzed in R using functions supported in the "Seurat" package unless otherwise described. To convert each file into a "Seurat" object, the "CreateSeuratObject" function was used. To enumerate the percent mitochondrial gene expression per cell per object, the "PercentageFeatureSet" function was used (pattern="^mt-"). To combine the objects into a single object, the "merge" function was used. The percent mitochondrial gene expression per cell along with the total number of detected counts and features per cell were then inspected using the "FeatureScatter" and "VlnPlot" functions. Cells were then filtered to keep only those that had: 1) a minimum number of detected counts > 500, 2) a maximum number of detected counts < observed mean + 2SD = 8916.1, 3) a minimum number of detected features > 250, 4) a maximum number of detected features < observed mean + 2SD = 3307.526, 5) a percent mitochondrial counts < observed mean + 1SD = 2.787142%. The "SplitObject" function was then used to separate surviving cells per library into a list where after the "as.SingleCellExperiment" function and the "computeDoubletDensity" function from the "scDbIFinder" package were applied. Cells observed to have a doublet score > respective library mean + 2SD were then filter removed from the pre-split object. The "Seurat::NormalizeData" function was then applied to this now doublet-curated object and the "FindVariableFeatures" function (selection.method = "vst", nfeatures = 3000) and the "ScaleData" function (vars.to.regress="percent.mt") sequentially applied. To determine the number of components to use in the clustering of cells, the "RunPCA" (npcs=200) and "ElbowPlot" (ndims) functions were used and the "RunPCA" function re-applied after using the number of components decided upon (npcs=75). To integrate cells across libraries, the "RunHarmony" (plot_convergence=TRUE) and "RunUMAP" (reduction="harmony", dims=1:75) functions were used. For clustering of cells, the "FindNeighbors" (reduction="harmony", dims=1:75) and "FindClusters" functions were used over a range of resolutions from 0.04 to 3.5 in steps of 0.01. Cluster results were summarized using the "clustree" package and 1.8 selected as the optimal resolution. At this resolution, several clusters were manually collapsed then the "findDoubletClusters" function from the "scDbIFinder" package used to screen for and remove doublet enriched clusters. The "computeDoubletDensity" function was then re-applied but this time without splitting the cells by library. Cells observed to have a doublet score > cross cell mean + 2SD were then filter removed and the "FindNeighbors" and "FindClusters" functions re-applied under the same conditions described. Cluster results were again summarized using the "clustree" package and 2.1 selected as the optimal resolution. At this resolution, several clusters were again manually collapsed then cells for each cluster z-scored using UMAP 1 dimension values and separately using UMAP dimension 2 values. Cells in z-score space were then plot inspected by cluster using the "DimPlot" function and thresholds defined and applied to filter remove cells having a z-score greater than the defined thresholds. Annotation of the resulting z-score curated clusters was accomplished using the "RunAzimuth" function from the "Azimuth" package in conjunction with the "mousecortexref.SeuratData" reference data set. To identify conserved markers across library class (WT and V321L), the "FindConservedMarkers" function was used. To identify nonconserved markers per cluster, the "FindMarkers" function was used. To identify differential features between library class within cluster, the "FindMarkers" function was used setting the "ident" parameters to "WT" and "V321L" respectively. Clusters annotated as "Dente gyrus" were next subset and trajectory fit using the "Monocle3" workflow. Root selection was accomplished by inspecting the expression for "Camk4" and "Ntng1" using the "plot_cells" function. The "graph_test" function (neighbor_graph="principal_graph") was used to identify features that explain pseudotime (q_value<0.05). To identify depleted vs enriched modules of features that explain pseudotime, the "find_gene_modules" function was used.

Single-nucleus ATAC-Seq Analysis: The "atac_peaks.bed", "barcode_metrics.csv", and "atac_fragments.tsv.gz" files produced for the "LL" and "VV" Dentate gyrus samples were imported into R (<https://cran.r-project.org/>) using the

"read.table" function. Functions available in the "Signac" and "Seurat" packages were interchangeably used for analysis. To create a working fragment object per sample, the "CreateFragmentObject" function was used. Cells with an "atac_fragments" value ≤ 500 were discarded with surviving cells used as input into the "FeatureMatrix", "CreateChromatinAssay", and "CreateSeuratObject" functions, producing one "Seurat" object per sample. For each of these objects, genomic annotations for "EnsDb.Mmusculus.v79" were added and quality metrics generated. To add the annotations, the "GetGRangesFromEnsDb" and "Annotation" functions were used. To generate the quality metrics, the "FractionCountsInRegion", "NucleosomeSignal", and "TSSEnrichment" functions were used. Metrics produced from these functions, along with those for "peak_region_fragments", "nCount", "pct_reads_in_peaks", and "TSS_fragments" were used to filter remove cells from each object via the "subset" function. Specifically, cells that did not have a value within 2SD of the observed mean for each metric were removed. Post filtering, all "Seurat" objects were combined into one using the "merge" function and a unified set of peaks generated using the "reduce" function. This unified peak set was then filtered to keep only those with a "peakwidths" value < 10000 and > 20 . The filtered peak set was then used to regenerate the individual "Seurat" objects as described and those objects combined into one object. The "RunTFIDF", "FindTopFeatures", "RunSVD", "RunUMAP", and "FindNeighbor" functions were then applied to the combined object (dims = 2:20, reduction = 'lsi') in preparation for clustering. For clustering, the "FindClusters" function was used over a range of resolutions from 0.04 to 3.5 in steps of 0.01. Cluster results were summarized using the "clustree" package and 1.5 selected as the optimal resolution. At this resolution, several clusters were manually collapsed then cells for each cluster z-scored using UMAP 1 dimension values and separately using UMAP dimension 2 values. Cells in z-score space were then plot inspected by cluster using the "DimPlot" function and filter thresholds defined. Cells having a z-score greater than the thresholds were removed. For surviving cells, annotations were transferred from the Single-nucleus RNA-Seq Analysis and log normalized activities for each known gene calculated. For annotation transfer, the "FindVariableFeatures" function (nfeatures = 5000) was used in conjunction with the "TransferData" function (reduction = 'cca', weight.reduction = 'lsi', dims = 2:20). To calculate the log normalized gene activities, and to add them to the combined "Seurat" object, the "GeneActivity", "CreateAssayObject", and "NormalizeData" functions were used. Specific clusters representing <HERE> were then subset and the functions used prior, per preparation for clustering, clustering, and post clustering, used again. For this subset, 0.5 was selected as the optimal resolution. Differential peaks between clusters produced at this resolution were tested for using the "FindMarkers" function (test.use = 'LR'). Overrepresented motifs available in the "JASPAR2020" database (collection = 'CORE', tax_group = 'vertebrates') were also tested for using the "FindMotifs" function. Post testing, results were annotated with the closest occurring known gene using the "ClosestFeature" function. In addition to this testing, trajectory fitting was performed using the "Monocle3" workflow. This workflow was applied in three separate facts: 1) using all cells representing "VV" and "LL" samples combined, 2) using cells for "VV" samples only, 3) using cells for "LL" samples only. Root selection for each of these facts was accomplished by inspecting the gene activities for "Camk4", "Igf1", "Fxyd7", "Dcx", "Calb2", "Calb1", "Ntn1", "Penk", and "Sorcs2" using the "plot_cells" function. Post fitting, the "cor.test" function was used in conjunction with the "p.adjust" function (method = 'BH') to test for and identify genes having pseudotime correlated with gene activity ($q_value < 0.05$). Enriched terms for these genes were subsequently identified using the "enrichR" package (dbs = 'GO_Molecular_Function_2023', 'GO_Cellular_Component_2023', 'GO_Biological_Process_2023'). Trajectory Geometry Analysis: Trajectory directionality was assessed using the TrajectoryGeometry R package. For each condition (WT downsampled, WT original, and V321L), PCA embeddings (top 50 principal components) and UMAP-derived pseudotime values were extracted from Monocle3 trajectory-fitted single-cell objects. PCA coordinates were rescaled to a 0–100 range to ensure comparability across datasets. The analyseSingleCellTrajectory() function was run with 1,000 sampling windows

(nSamples = 1000) and 1,000 randomized path permutations (N = 1000) using the randomizationParams = c("byPermutation", "permuteWithinColumns") setting to assess statistical significance of trajectory directionality based on spherical distances. The test statistic was set to "mean" to capture average directional consistency."

We hope this additional information satisfies the reviewer's request and enhances clarity regarding our bioinformatics analyses.

4. The authors argue that the existence of distinct differentiation trajectories leading to terminal GC types supports the idea that SGCs represent a unique cell type rather than a terminal state that any GC can achieve. However, the cell clusters traversed in WT and V321L mice are not balanced across donors. A downsampling strategy could help ensure a more accurate comparison of lineage reconstructions.

We thank the reviewer for raising this important point regarding potential sampling bias between genotypes, which could impact clustering and the validity of inferred differentiation trajectories. To address this concern, we implemented two complementary strategies:

- **Downsampling Approach:**

As suggested, we performed downsampling to equalize cell numbers across WT and V₃₂₁L conditions and re-evaluated pseudotime trajectories and terminal trajectory-associated gene expression.

- **TrajectoryGeometry Analysis (Laddach, Pachnis et al. 2024):**

This method samples cells over 1000 iterations, either in pseudotime order (as predicted) or in a randomized order, using equal cell numbers across conditions. It evaluates the directional consistency of gene expression dynamics in an N-dimensional space by projecting trajectory endpoints onto an N-1 dimensional sphere.

Lower spherical distances indicate more directed (mono-directional) trajectories.

Higher distances suggest branching complexity or fluctuating gene expression dynamics.

Key Findings:

- **Downsampling** did not alter major conclusions. The maximal pseudotime and associated clusters for SGCs remained consistent. Additionally, 90% of the top 100 pseudotime-associated genes were conserved between the original and downsampled datasets, with Sorcs3 and Nrg1 consistently ranking as the top genes explaining pseudotime progression.
- **TrajectoryGeometry** revealed that predicted trajectories were significantly more directed than randomized controls in all datasets (WT original, V₃₂₁L, and WT downsampled), supporting their biological relevance.

WT trajectories showed greater spherical distances than V₃₂₁L, indicating more complex lineage branching in WT. This suggests that V₃₂₁L differentiation follows a more simplified, unidirectional path.

Notably, even when using all available WT cells, the statistical distance remained higher than in V₃₂₁L, implying that more WT cells may be necessary to reach equivalent statistical confidence. This highlights the importance of sample size when interpreting trajectories with greater complexity.

Furthermore, our ATAC-Seq trajectory analysis reinforces these findings. Despite balanced sampling, the chromatin accessibility trajectories in V₃₂₁L displayed a single, unidirectional path, in contrast to the

more branched WT pattern. This supports the interpretation that lineage complexity varies between genotypes.

Importantly, we would like to emphasize that trajectory inference is not absolute or exhaustive, but relative to the data used for modeling. A significant trajectory fit in the downsampled space reflects the structure present in that specific subset, while the full dataset may reveal additional complexity. These results should be interpreted as valid approximations of lineage progression given the sampled content, rather than definitive maps of differentiation. We have clarified this point in the revised manuscript text.

We have now incorporated the downsampled pseudotime gene expression data in Table S2 and added a **new supplemental figure (Figure S7)** showing the UMAP of the downsampled pseudotime analysis. We have also updated the Results and Discussion sections to reflect these findings and their implications for understanding genotype-specific lineage complexity.

Lines 381-402: *“To address differences in cell sampling between WT and V₃₂₁L GCs, we performed a down sampling of WT GCs and recalculated pseudotime and gene expression associated with pseudotime (Figure S7A). We found that the pseudotime ordering of GCs and associated gene expression states were robust to differences in sampling frequency, with Sorcs3 still representing cells with the maximal pseudotime (Figure S7B). We next used “Trajectory Geometry” to assess whether the identified trajectories in WT and V₃₂₁L mice have well defined directionality (Laddach, Pachnis et al. 2024). First, we tested cells along the identified trajectories in the WT mice (original and downsampled) against 1000 iterations of pseudotime randomized cell trajectories. WT original and down sampled trajectories resulted in significantly lower spherical distances compared to randomized ones indicating the presence of a definite directionality within the trajectories (Wilcoxon Signed Rank Test- Randomized vs. WT original $p=4.25*10^{-11}$, Randomized vs. WT down sampled $p=8.00*10^{-31}$). Similarly, the V₃₂₁L trajectory also produced a significantly smaller spherical distance relative to randomized trajectories (Wilcoxon Signed Rank Test- Randomized vs. V₃₂₁L $p=3.17*10^{-54}$). Interestingly, the spherical distances obtained for the WT trajectories were greater than those for the V₃₂₁L trajectory indicating the presence of more complex fate decisions as seen by the branching in the WT trajectory. Importantly, while both the original and downsampled WT data yielded valid and statistically significant trajectories, they reflect different levels of complexity based on sampling. Thus, trajectory inference is not exhaustive but rather an approximation of lineage dynamics based on available cellular content.”*

Lines 628-635: *“Furthermore, Trajectory Geometry analyses revealed that the V₃₂₁L GCs take a more directed trajectory potentially reflecting a loss of complexity in lineage decisions thereby reducing the overall heterogeneity of the population.*

Given the unique circuit properties of SGCs and their potential to release enkephalin (encoded by Penk), we suggest that interactions between developmentally regulated signaling and chromatin modifying enzymes might afford the DG some flexibility to regulate cell type composition depending on the context of the circuit, to maintain proper circuit dynamics (Larimer and Strowbridge 2010, Gupta, Proddutur et al. 2020).”

5. The proposed involvement of PRC2 in repressing the SGC phenotype is not well substantiated. More details on the ChEA/ENCODE analysis or alternative approaches should be provided. For instance, a permutation-based approach could assess the likelihood of PRC2 targets being enriched in random sets of GC-expressed genes of similar size (e.g., 60 genes).

The ChEA/ENCODE analysis was conducted using Enrichr, where transcriptional regulators were identified using Fisher’s exact test, with Benjamini-Hochberg correction applied to account for multiple testing across 104 transcriptional regulator terms covering over 15,000 genes. A significance threshold of adjusted $p < 0.05$ was used to filter results, leaving four enriched transcriptional regulators. Among these, Suz12—a core component of PRC2—had the smallest adjusted p-value and regulated the highest number of genes compared to the other candidates.

Furthermore, our previous study demonstrated that core PRC2 components were downregulated in the V₃₂₁L DG, providing additional support for its potential role in regulating SGC fate (Rajebhosale, Jone et al. 2024).

However, in response to the reviewer's point, we conducted an additional analysis to assess transcriptional regulators enriched at alternative "differentiation" endpoints. Specifically, we analyzed Cluster 5, which was devoid of *Penk*⁺ GCs in WT mice and represented a distinct endpoint diverging from the SGC trajectory. We identified 21 transcriptional regulators with an adjusted $p < 0.05$ that were enriched in genes marking end trajectories and Cluster 5 markers. None of these 21 regulators belonged to PRC2, suggesting that PRC2 activity may be more specifically associated with the SGC trajectory rather than alternative differentiation fates.

To enhance transparency, we have now included this list of candidate regulators, their predicted target genes, and full statistical details in **Table S3**.

6. Investigating recently born SGCs to identify signatures unrelated to cell maturation is a valuable approach. The authors describe *Penk* expression and other SGC-enriched genes, but it would be informative to examine the expression of *Sorcs3* and *Nptx2* in these early SGCs as well.

We initially shared the reviewer's concern regarding differentiating effects of cell maturation vs. cell type signatures. **Lines 453-461** along with **New Figure S9C** attempted to address this concern.

"Since *Penk*⁺ GCs are embryonically generated and bulk of the GCs are not, we wondered if the SGC-like transcriptome might simply reflect cellular age and not necessarily mark SGCs. We examined ISH data from embryonic day 18.5 (E18.5) mouse DG (Allen Brain Atlas), when the newborn SGCs would be between 0 and 3 days of age, for expression of a subset of the SGC-enriched genes from our snRNASeq analysis (**Figure S9C**). We found cells expressing SGC-enriched genes such as *Penk*, *Scg2*, *Nrg1* and *Cit* in the dorsal border of the developing GCL at E18.5 indicating that marker gene expression is unlikely to be an artifact of cellular age (**Figure S9C**)."

Additionally, the difference in birthdates between SGCs and the bulk GC population is at most one week. Given that most of our marker gene analyses were conducted at 2–3 months of age, if these markers were simply indicative of cellular age, we would expect a much larger proportion of GCs to express these genes over time. However, our data reveal that SGC marker-expressing cells remain a rare subpopulation throughout early life, maintaining their distinct identity even after a modest expansion during adolescence.

Minor comments:

In Figure 2A, some boxes of the boxplot are colored black, and medians cannot be distinguished.

This has been corrected.

References

Auffenberg, E., A. Jurik, C. Mattusch, R. Stoffel, A. Genewsky, C. Namendorf, R. M. Schmid, G. Rammes, M. Biel, M. Uhr, S. Moosmang, S. Michalakakis, C. T. Wotjak and C. K. Thoeringer (2016). "Remote and reversible inhibition of neurons and circuits by small molecule induced potassium channel stabilization." Sci Rep **6**: 19293.

Fitzsimons, C. P., L. W. van Hooijdonk, M. Schouten, I. Zalachoras, V. Brinks, T. Zheng, T. G. Schouten, D. J. Saaltink, T. Dijkmans, D. A. Steindler, J. Verhaagen, F. J. Verbeek, P. J. Lucassen, E. R. de Kloet, O. C. Meijer, H. Karst, M. Joels, M. S. Oitzl and E. Vreugdenhil (2013). "Knockdown of the glucocorticoid receptor alters functional integration of newborn neurons in the adult hippocampus and impairs fear-motivated behavior." Mol Psychiatry **18**(9): 993-1005.

Kim, J. Y., X. Duan, C. Y. Liu, M. H. Jang, J. U. Guo, N. Pow-anpongkul, E. Kang, H. Song and G. L. Ming (2009). "DISC1 regulates new neuron development in the adult brain via modulation of AKT-mTOR signaling through KIAA1212." Neuron **63**(6): 761-773.

Laddach, A., V. Pachnis and M. Shapiro (2024). "TrajectoryGeometry suggests cell fate decisions can involve branches rather than bifurcations." NAR Genom Bioinform **6**(4): lqae139.

Lauer, M., H. Beckmann and D. Senitz (2003). "Increased frequency of dentate granule cells with basal dendrites in the hippocampal formation of schizophrenics." Psychiatry Res **122**(2): 89-97.

Llorens-Martin, M., J. Jurado-Arjona, A. Fuster-Matanzo, F. Hernandez, A. Rabano and J. Avila (2014). "Peripherally triggered and GSK-3beta-driven brain inflammation differentially skew adult hippocampal neurogenesis, behavioral pattern separation and microglial activation in response to ibuprofen." Transl Psychiatry **4**(10): e463.

Marquez-Valadez, B., A. Rabano and M. Llorens-Martin (2022). "Progression of Alzheimer's disease parallels unusual structural plasticity of human dentate granule cells." Acta Neuropathol Commun **10**(1): 125.

Rajebhosale, P., A. Jone, K. R. Johnson, R. Hofland, C. Palarpalar, S. Khan, L. W. Role and D. A. Talmage (2024). "Neuregulin1 Nuclear Signaling Influences Adult Neurogenesis and Regulates a Schizophrenia Susceptibility Gene Network within the Mouse Dentate Gyrus." J Neurosci **44**(43).

Terreros-Roncal, J., M. Flor-Garcia, E. P. Moreno-Jimenez, N. Pallas-Bazarra, A. Rabano, N. Sah, H. van Praag, D. Giacomini, A. F. Schinder, J. Avila and M. Llorens-Martin (2019). "Activity-Dependent Reconnection of Adult-Born Dentate Granule Cells in a Mouse Model of Frontotemporal Dementia." J Neurosci **39**(29): 5794-5815.

April 11, 2025

RE: Life Science Alliance Manuscript #LSA-2024-03169-TR

Dr. David A Talmage
National Institute of Neurological Disorders and Stroke
35 Convent Ave
3B 1012
Bethesda, MD 20892

Dear Dr. Talmage,

Thank you for submitting your revised manuscript entitled "Diversification of Dentate Gyrus Granule Cell Subtypes is Regulated by Nrg1 Nuclear Back Signaling". We would be happy to publish your paper in Life Science Alliance pending final revisions necessary to meet our formatting guidelines.

- please be sure that the authorship listing and order is correct.
- please add the X and Bluesky handles of your host institute/organization as well as your own or/and one of the authors in our system.
- please note that titles in the system and manuscript file must match.
- please use the [10 author names, et al.] format in your references (i.e., limit the author names to the first 10).
- please add a callout for Figure fig. 5E and Tables S5, S6 to your main manuscript text.

A. FINAL FILES:

B. MANUSCRIPT ORGANIZATION AND FORMATTING:

Sincerely,

Reviewer #1 (Comments to the Authors (Required)):

In the study "Diversification of Dentate Gyrus Granule Cell Subtypes is Regulated by Neuregulin1 Nuclear Back Signaling," Rajebhosale and colleagues focus on two types of GCs (typical and semilunar), providing a detailed characterization of these cell types, and how their abundance, electrophysiology, and spatial location change with a disease-associated genetic mutation (V321L). They then provide a plausible mechanism for their findings and show that similar changes occur in normal aging. This study advances previous work by showing that GCs can directly transition into SGCs and that this transition may be suppressed by nuclear back propagation, which has previously been associated with adult neurogenesis. Furthermore, the multimodal characterization of these cell subtypes with morphology, electrophysiology, transcriptomics, epigenomics, and spatial profiling in both healthy and disease mice is extensive.

This is a well-written paper that clearly describes the problem being addressed in the context of what is currently known about GC development, and then presents a series of experiments and associated analysis to compellingly address the problem.

In the initial revision, all reviewer requests were addressed. This work presents an advance to the field, and no additional revisions are suggested.

Reviewer #2 (Comments to the Authors (Required)):

I appreciate the authors' efforts in satisfactorily addressing my concerns, and I recommend the paper for publication.

April 16, 2025

RE: Life Science Alliance Manuscript #LSA-2024-03169-TRR

Dr. David A Talmage
National Institute of Neurological Disorders and Stroke
35 Convent Ave
3B 1012
Bethesda, MD 20892

Dear Dr. Talmage,

Thank you for submitting your Research Article entitled "Diversification of Dentate Gyrus Granule Cell Subtypes is Regulated by Nrg1 Nuclear Back Signaling". It is a pleasure to let you know that your manuscript is now accepted for publication in Life Science Alliance. Congratulations on this interesting work.

DISTRIBUTION OF MATERIALS:

Again, congratulations on a very nice paper. I hope you found the review process to be constructive and are pleased with how the manuscript was handled editorially. We look forward to future exciting submissions from your lab.

Sincerely,
